# Rethinking Fine-Tuning when Scaling Test-Time Compute: Limiting Confidence Improves Mathematical Reasoning

**Feng Chen**[*]
Stanford University
Stanford, CA 94305
fengc@stanford.edu

**Allan Raventós**[*]
Stanford University
Stanford, CA 94305
aravento@stanford.edu

**Nan Cheng**
University of Michigan
Ann Arbor, MI 48109
nancheng@umich.edu

**Surya Ganguli**
Stanford University
Stanford, CA 94305
sganguli@stanford.edu

**Shaul Druckmann**
Stanford University
Stanford, CA 94305
shauld@stanford.edu

## Abstract

Recent progress in large language models (LLMs) highlights the power of scaling test-time compute to achieve strong performance on complex tasks, such as mathematical reasoning and code generation. This raises a critical question: how should model training be modified to optimize performance under a subsequent test-time compute strategy and budget? To explore this, we focus on pass@N, a simple test-time strategy that searches for a correct answer in $N$ independent samples. We show, surprisingly, that training with cross-entropy (CE) loss can be *misaligned* with pass@N in that pass@N accuracy *decreases* with longer training. We explain the origins of this misalignment in terms of model overconfidence induced by CE, and experimentally verify our prediction of overconfidence as an impediment to scaling test-time compute via pass@N. Furthermore we suggest a principled, modified training loss that is better aligned to pass@N by limiting model confidence and rescuing pass@N test performance. Our algorithm demonstrates improved mathematical reasoning on MATH and MiniF2F benchmarks under several scenarios: (1) providing answers to math questions; and (2) proving theorems by searching over proof trees of varying shapes. Overall our work underscores the importance of co-designing two traditionally separate phases of LLM development: training-time protocols and test-time search and reasoning strategies.

## 1 Introduction

Scaling test-time compute has been integral to unprecedented improvements in LLMs' reasoning skills for complex tasks such as math and coding. Thus, test-time compute has emerged as a new dimension for improving LLMs, leading to a key tradeoff between allocating additional compute to inference versus pretraining [1]. Diverse test-time strategies include Chain-of-Thought (CoT) [2], tree-of-thought [3], self-consistency [4], self-reflection [5], self-critique [6], self-verification [7] and Monte-Carlo tree search [8]. These have shown great success in boosting model performance in the post-training phase or at inference time. More recently, OpenAI's O1 model [9] and DeepSeek's R1 model [10] have combined some of these strategies with reinforcement learning to generate high-

---

[*]Equal Contribution

39th Conference on Neural Information Processing Systems (NeurIPS 2025).

quality reasoning traces for problems of various difficulty levels, demonstrating clear performance improvements as more test-time compute is allocated.

These successes fit into a broader paradigm in which a frontier model is first fine-tuned on a reasoning task with supervised fine-tuning (SFT) [2, 11, 12], and then a test-time algorithm is applied to model outputs or reasoning traces to improve performance [3, 4, 13]. Many test-time algorithms are independent of the fine-tuning process. As a result, the fine-tuning is agnostic to and thus decoupled from the test-time algorithm [14]. However, for a given choice of test-time strategy and compute budget, it is not *a priori* clear which fine-tuning approach, including the loss objective, would be best aligned with the test-time strategy so as to maximize the test accuracy under the overall strategy.

Our work studies the problem of aligning fine-tuning with test-time algorithms. We consider what is perhaps the simplest setting, SFT with CE loss and pass@N as the test-time strategy. This setting reveals a case of misalignment: standard SFT is not maximizing performance under pass@N. We believe that this misalignment presents itself in several combinations of fine-tuning/test-time approaches, motivating our thorough study in this paper. Our main contributions are,

- We identify a misalignment between standard fine-tuning with CE loss and the pass@N coverage metric at test time (Section 4.1).

- We develop and experimentally verify a framework that suggests this misalignment arises from overconfidence induced by training on CE loss (Sections 4.2 and 4.3).

- We propose a loss function that directly optimizes the pass@N coverage metric, demonstrating consistent improvement over the CE loss objective and achieving superior accuracy frontiers on MATH and MiniF2F (Sections 5.1 to 5.3).

- We extend our algorithm to more complex test-time scenarios including searching over proof-trees of varying shapes to improve automated theorem proving, and answering math questions using Chain-of-Thought reasoning traces, demonstrating improved mathematical reasoning in both cases (Sections 5.3 and 5.4).

## 2 Related Works

**Test-time compute and pass@N strategy.** A growing body of work explores how training and test-time strategies interact to shape model performance. Jones [15] highlight a tradeoff between train and test-time compute in a controlled board game setting. Brown et al. [16] identify an exponentiated power law between coverage and number of in-context samples, captured phenomenologically in Levi [17] and Schaeffer et al. [18], while Snell et al. [1] explore compute-optimal strategies for scaling test-time inference. Our paper focuses primarily on the pass@N test-time strategy. Gui et al. [19] show that, in the best-of-N setting, the sample distribution is well-aligned with the reward model, and alignment methods have been proposed to distill this distribution and reduce sampling costs [19, 20]. However, their work focuses on improving the pass@1 performance from the Best-of-N policy, while we focus on directly improving the pass@N performance. Closely related to our work, Chow et al. [14] derives a similar training objective for reinforcement learning (RL) to directly optimize for the best-of-N test-time strategy. Li et al. [21] argue that fine-tuning with cross-entropy loss limits output diversity and propose a maximum entropy-based method to address this issue. Our work similarly identifies limitations of cross-entropy when scaling test-time compute, but focuses on the alignment of training and test-time strategies. In addition, Yue et al. [22] argue that fine-tuning with Group Relative Policy Optimization (GRPO) does not improve reasoning capabilities beyond the base model when evaluated with pass@N strategy. We will show that GRPO training may also lead to overconfidence in Figure 3.

**Post-training for mathematical reasoning.** Various post-training techniques have been developed to improve mathematical reasoning in LLMs. Instruction-tuning and reinforcement learning with human feedback boost model performance on math tasks [23–25], while continued training on domain-specific math or code data also boosts downstream reasoning [26–30]. Rejection-sampling [31] and self-improvement [32] methods augment training data for SFT. Recent approaches [9, 10] have incorporated RL to achieve exceptional reasoning capabilities. GRPO [10, 29] is a newly proposed RL algorithm demonstrating strong improvements in mathematical reasoning. Several recent studies have further addressed response length and problem difficulty biases [33], as well as conducted extensive analyses of accuracy and response length across a wide variety of base models [33, 34].

While our primary focus is SFT, our loss function can be applied to other settings that train under CE loss, such as continual training, instruction-tuning, and data augmentation.

**Data pruning and hard example mining.** The loss function we derive below can be viewed from the lens of data pruning and hard example mining. Data selection is used to curate high-quality datasets for pretraining [35], where Sorscher et al. [36] show that pruning easy samples can improve loss scaling as a function of dataset size. Zhou et al. [37], Ye et al. [38] demonstrate that, in SFT, even small datasets can yield strong alignment and reasoning performance. Hard example mining focuses on identifying and emphasizing challenging samples to improve model performance [39]. In the domain of mathematical reasoning, Tong et al. [40] find a difficulty imbalance in rejection-sampled datasets and show that more extensive training on difficult samples improves model performance.

## 3    Problem setup

Given a vocabulary set $W$, we consider a dataset $\mathcal{D} = \{(x^{(i)}, y^{(i)})\}_{i=1}^{M}$, where $x^{(i)} \in W^{n_i}$ is a prompt, $y^{(i)} \in W^{m_i}$ is its ground-truth completion, and $n_i$ and $m_i$ are the prompt and completion lengths. In the context of math, $x^{(i)}$ is the problem statement and $y^{(i)}$ is its solution. To model the conditional distribution $p(y^{(i)}|x^{(i)})$ we use an autoregressive transformer model [41], which is traditionally trained by minimizing the cross-entropy loss

$$\mathcal{L}_{\text{CE}} = - \mathop{\mathbb{E}}_{(x,y) \sim \mathcal{D}} \log \hat{p}(y|x) \tag{1}$$

where $\hat{p}$ denotes the model's distribution.

To use and evaluate the model at test time, we assume the existence of an efficient oracle verifier $V$ which takes as input an $(x, y)$ pair and returns $V(x, y) = 1$ if $y$ is a correct completion of $x$ and $0$ otherwise. Practical examples of verifiers include compilers or pre-defined unit tests for coding problems, or automatic proof checkers in mathematical theorem proving. In such applications, a simple method, known as pass@N, for trading test-time compute for accuracy involves sampling $N$ completions from $\hat{p}$ given the test prompt $x$ and applying the verifier $V$ to all of them to search for a correct solution. The probability of a correct answer is then no longer the probability that $1$ completion is correct, but rather the probability that *at least one* of $N$ is correct. This probability, for a dataset $\mathcal{D}$, is given by the pass@N coverage metric

$$\mathcal{C}_{\mathcal{D}}^{N} = \mathop{\mathbb{E}}_{x \sim \mathcal{D}, y_i \overset{\text{i.i.d.}}{\sim} \hat{p}(\cdot|x)} \mathbb{P}(\exists j \in [N] \, s.t. \, V(x, y_j) = 1). \tag{2}$$

Minimizing the CE loss in Equation (1) is equivalent to maximizing the pass@1 metric $\mathcal{C}_{\mathcal{D}}^{1}$ on a training set $\mathcal{D}$. But if we scale up test-time compute so that the pass@N metric $\mathcal{C}_{\mathcal{D}}^{N}$ on a test set $\mathcal{D}$ for $N \gg 1$ is the relevant performance metric, is $\mathcal{L}_{\text{CE}}$ still a good training loss, or can we do better?

## 4    Misalignment between CE loss and pass@N

### 4.1    The CE loss induces overfitting for pass@N

To understand the impact of training with CE loss on pass@N test performance, we fine-tune Llama-3-8B-base [42] on the MATH [43] dataset. We start from the base model rather than Llama-3-8B-Instruct to avoid potential leakage of the MATH dataset into Llama-3-8B-Instruct through post-training. We follow Lightman et al. [24] and use $12,000$ problems for training and the remaining $500$ for testing. Here we train the model to provide a direct answer without a reasoning trace. We will discuss training with CoT in Section 5.4. We confirm that the results presented in this section also hold for a larger model, Llama-3-70B-base, and direct readers to Appendix C.3 for additional results.

Table 1 reveals that the pass@N performance $\mathcal{C}_{\mathcal{D}}^{N}$ on a test set monotonically increases with the number of training epochs *only* for $N = 1$, when minimizing CE loss is equivalent to maximizing $\mathcal{C}_{\mathcal{D}}^{1}$. However, for $N \geq 16$, minimizing CE loss during training does *not* monotonically increase $\mathcal{C}_{\mathcal{D}}^{N}$ at test; indeed for $N \geq 256$, pass@N test performance, remarkably, monotonically *decreases* with the number of training epochs, despite the fact that pass@1 performance monotonically *increases*. This effectively corresponds to a novel type of overfitting, in which test performance degrades over training, likely due to a mismatch between the test time (pass@N) and training time (pass@1) strategies.

## 4.2 Overfitting, confidence, and explore-exploit tradeoff

What are the origins of this overfitting? First we show that in the simple case of a *single* problem $x$, overfitting cannot occur. Let $\hat{p}(x)$ denote the probability assigned by the model to all correct answers for problem $x$. Then the pass@1 coverage is $\mathcal{C}^1 = \hat{p}(x)$ while the pass@N coverage is $\mathcal{C}^N = 1 - (1 - \hat{p}(x))^N = 1 - (1 - \mathcal{C}^1)^N$. This formula for $\mathcal{C}^N$ in *the single problem* case obeys:

**Lemma 4.1.** $\forall N, N' > 0$, $\mathcal{C}^N$ is monotonic in $\mathcal{C}^{N'}$.

Thus increasing pass@N$'$ coverage implies increasing pass@N coverage for any $N$ and $N'$. When $N' = 1$, this implies minimizing CE loss maximizes pass@N coverage.

However, Lemma 4.1 can fail when there is more than one problem. To understand this in a simple setting, consider a test set with two problems $x_1$ and $x_2$ with unique correct answers $y_1$ and $y_2$. Consider two models. The first model assigns probabilities $\hat{p}_1(y_1|x_1) = 1$ and $\hat{p}_1(y_2|x_2) = 0$. If we think of the probability a model assigns to an answer as its *confidence* in that answer, this model is highly confident and correct for problem $x_1$, but highly confident and *wrong* for $x_2$. Its pass@1 coverage is thus 50%. Moreover, since it always gets $x_1$ right and $x_2$ wrong, its pass@N coverage remains at 50%. In contrast, consider a second model which assigns probabilities $\hat{p}_2(y_1|x_1) = \hat{p}_2(y_2|x_2) = 0.1$. This model has high confidence (0.9) but unfortunately on incorrect answers. Therefore its pass@1 accuracy is only 10%. However, it is willing to explore or hedge by placing some low confidence (0.1) on the other answer, which happens to be correct. Thus if this model samples $N$ times, the pass@N coverage increases with $N$, eventually approaching 100% as $N \to \infty$. Thus the first model outperforms the second in terms of pass@1 but not pass@N for large $N$. This indicates a tradeoff amongst policies: those that do better on pass@1 may not do better at pass@N. Of course the best one can do is be confident *and* correct, corresponding to the optimal model with $\hat{p}^*(y_1|x_1) = \hat{p}^*(y_2|x_2) = 1$. However, this toy example reveals that if one cannot guarantee correctness, it can be beneficial to limit confidence and explore more solutions.

To demonstrate more generally the existence of tradeoffs between confident exploitation of a few answers versus unconfident exploration of many answers, as a function of the number of passes $N$, we prove two lemmas. To set up these lemmas, given any problem, let $\hat{p}_i$ denote the model probability or confidence assigned to answer $i$, and assume answers are sorted from highest to lowest confidence so that $\hat{p}_i \geq \hat{p}_{i+1}, \forall i \geq 1$. Thus $\hat{p}_1$ is the model's *maximal confidence* across all answers. Moreover, let $p_i$ be the probability across all the problems that the $i^{\text{th}}$ ranked answer is *actually* correct. Assume the model policy is approximately well calibrated so that higher confidence implies higher or equal probability of being correct, i.e. $p_i \geq p_{i+1}, \forall i \geq 1$. We empirically verify that the model trained with CE loss on the MATH dataset approximately satisfies this assumption (Figure 13). Indeed, optimal policies maximizing pass@N coverage in Equation (2) are approximately well calibrated (See Lemma A.1). $p_1$ is then the model's *maximal accuracy* across all answers. We prove:

**Lemma 4.2** (Upper bound on max confidence). *Assume a max accuracy $p_1$, and assume $\sum_{i=1}^{k} p_i \geq 1 - \epsilon$ for some $0 < \epsilon < 1 - p_1$. Then optimal policies maximizing pass@N coverage in Equation (2), subject to the above accuracy and calibration constraints, have a confidence upper bound $u(N, p_1, \epsilon, k)$, which is a non-increasing function of $N$.*

We proved this upper bound as well as the proof in Appendix A. This upper bound is monotonically decreasing in $N$ (Figure 12), implying that at large $N$, an optimal policy for pass@N must limit its

Table 1: Pass@N coverage on the MATH test set for a Llama-3-8B-base model fine-tuned with CE loss on direct answers from the MATH training set. While pass@1 improves with continued training, for large $N$, surprisingly, pass@N decreases with number of training epochs.

|         | pass@1 | pass@16 | pass@256 | pass@4k |
|---------|--------|---------|----------|---------|
| Epoch 1 | 4.4%   | 30.0%   | **65.2%**| **82.5%**|
| Epoch 2 | 5.3%   | **31.4%**| 64.5%   | 80.0%   |
| Epoch 3 | 6.5%   | 28.7%   | 54.5%    | 79.2%   |
| Epoch 4 | **7.4%**| 22.9%  | 44.5%    | 63.0%   |

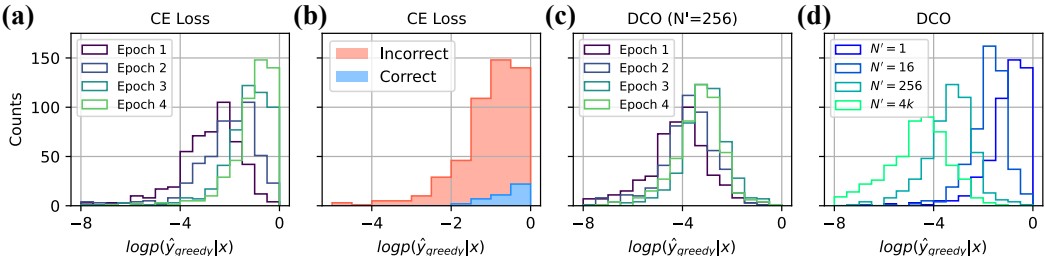

Figure 1: **A model trained with CE loss becomes overconfident in its greedy completions, which harms its pass@N coverage; our proposed DCO objective limits this overconfidence.** We fine-tune a Llama-3-8B base model on the MATH dataset to produce direct answers. $\hat{y}_{\text{greedy}}$ is the model's greedy completion when choosing the most likely token at each sampling step. **(a)** The model trained with CE loss assigns progressively larger confidences $\hat{p}(\hat{y}_{\text{greedy}}|x)$ to its greedy completions over the course of training. **(b)** At the end of training, only a small portion of the model's highly confident completions are correct. This will harm the model's pass@N performance when scaling up $N$. **(c)** Same as (a) but shown for the DCO loss with $N' = 256$. The model trained on DCO shows a much milder overconfidence effect. **(d)** The confidence distribution of greedy completions after four epochs with DCO for various choices of $N'$. As $N'$ increases, the model's confidence in its greedy completion is more stringently limited, directly as a consequence of the overconfidence regularizer $F$.

max confidence to be low, thereby favoring exploration and discouraging exploitation. Furthermore, we prove:

**Lemma 4.3** (Lower bound on max confidence). *Assume top two accuracies $p_1 > p_2$. Then optimal policies maximizing pass@N coverage in Equation* (2)*, subject to above accuracy and calibration constraints, must have max confidence obeying*

$$\hat{p}_1^* \geq 1 - \frac{(1 - p_1 + p_2)p_1^{\frac{1}{1-N}} p_2^{\frac{1}{N-1}}}{1 - p_1 + p_2 + p_1^{\frac{1}{1-N}} p_2^{\frac{N}{N-1}}}.$$

This lower bound is a monotonically decreasing function of $N$ (Figure 12), implying that at small $N$, optimal policies must have high max confidence, thereby favoring exploitation and discouraging exploration. Note in the limit $N \to 1^+$, the lower bound is always 1, recovering the intuitive result that the optimal policy for pass@1 is to place 100% confidence on the highest accuracy answer.

### 4.3 Overconfidence prevents performance gains from scaling test-time compute

Lemma 4.2 suggests that at large $N$ it is beneficial to unconfidently explore by assigning low model confidence to many answers, while Lemma 4.3 suggests that at small $N$ it is beneficial to confidently exploit by assigning high model confidence to one or a few answers. These lemmas make a prediction that could explain the empirical observation in Table 1: namely, maximization of pass@1 makes the model overconfident, thereby preventing good performance on pass@N.

To test this theoretical prediction of model overconfidence, we estimated the max confidence of the model as $\hat{p}(y_{\text{greedy}}|x)$ where $y_{\text{greedy}}$ is the greedy completion to $x$ obtained by sampling autoregressively from the model $\hat{p}$, and at each step selecting the token with the highest probability. $\hat{p}(y_{\text{greedy}}|x)$ approximates the max confidence $\hat{p}_1$ in Lemmas 4.2 and 4.3. For the model fine-tuned on MATH with CE loss, we plot the distribution of $\hat{p}(y_{\text{greedy}}|x)$ over the test set in Figure 1 (a). This demonstrates the model becomes progressively more confident about its greedy completion over training, thereby confirming our theoretical prediction. In Figure 1 (b) we see why this is a problem: only some of the model's highly confident answers are correct. Thus the model becomes overconfident *and* largely wrong. Scaling test-time compute cannot easily rescue such a model, as over multiple samples, it is likely to confidently provide the same wrong answers, explaining the origins of poor pass@N test performance. In Appendix C.4, we observe a similar trend of overconfidence on the out-of-distribution test set AIME24, further highlighting the generality of this failure mode.

# 5 Direct Coverage Optimization

## 5.1 A solution that naturally prevents overconfidence

The misalignment between CE training loss and pass@N coverage suggests a simple solution: *directly optimize* pass@N coverage for each training example at training time, whenever the pass@N strategy is to be used at test-time. We thus propose the Direct Coverage Optimization (DCO) objective, $\mathcal{L}_{\text{DCO}}^N = \mathbb{E}_{(x,y)\sim\mathcal{D}} \ell_{\text{DCO}}^N(x,y)$, where

$$\ell_{\text{DCO}}^N(x,y) = -\log\left(1 - (1 - \hat{p}(y|x))^N\right), \tag{3}$$

is the $-\log$ probability that the model produces $y$ at least once in $N$ samples given $x$. Thus for each prompt $x$ and correct completion $y$ in the training set, we maximize the probability $y$ is found in $N$ passes. This loss naturally prevents model overconfidence, as can be seen via its gradient

$$\nabla_\theta \ell_{\text{DCO}}^N(x,y) = F\left(N, \hat{p}(y|x)\right) \nabla_\theta \ell_{\text{CE}}(x,y) \tag{4}$$

where $\ell_{\text{CE}}$ is the standard CE loss on a single example, and $F(N, \hat{p}(y|x)) = \frac{N(1-\hat{p}(y|x))^{N-1}\hat{p}(y|x)}{1-(1-\hat{p}(y|x))^N}$ is an extra overconfidence regularization factor that multiplies the standard CE gradient. Note that $F(N, \hat{p}(y|x)) = 1$ for $N = 1$, so DCO reduces to CE for pass@1. Furthermore $F(N, \hat{p}(y|x))$ monotonically decreases in the model confidence $\hat{p}(y|x)$ (see Figure 2 (b)). Thus gradients for examples $(x, y)$ on which the model is more confident are attenuated, and this attenuation is stronger for larger $N$. This justifies the interpretation of $F(N, \hat{p}(y|x))$ as a regularizer that prevents model overconfidence. Indeed, for large $N$, $F(N, \hat{p}(y|x)) \approx 0$ for confidence $\hat{p}(y|x) \gtrsim 1/N$. As soon as model confidence on an example exceeds $1/N$, its gradient becomes negligible. Thus interestingly, aligning training and test better through DCO naturally yields a simple emergent regularization of model overconfidence, which was itself identified as an impediment to performance gains through scaling test-time compute in Figure 1 (a). Indeed, $F(N, \hat{p}(y|x))$ is smaller for easier examples (Figure 14, right), effectively regularizing their contribution. This regularization mitigates the potential detrimental effects of overconfidence from easy examples as discussed in Appendix C.1.

In practice, the introduction of $F$ can lead to some samples in a batch contributing minimally to the current gradient step. To maintain a stable effective batch size, we introduce a threshold $\epsilon$, and if for a given example, $(x, y)$, $F(N, \hat{p}(y|x)) < \epsilon$, we replace the example with a new one.

## 5.2 DCO can prevent overconfidence and rescue test-time scaling

We next test whether DCO can rescue test-time scaling by preventing model overconfidence. We first perform experiments on the MATH dataset in this section, and then in the next section we perform experiments on the LeanDojo automated theorem proving benchmark [44]. We fine-tune Llama-3-8B base model for 4 epochs on the MATH training set using DCO for $N' = 256$ and confirm that the model is much less confident in its greedy completion than when trained with CE loss after multiple epochs (compare Figure 1 (a) and (c)). Moreover, training with DCO at larger $N$ yields lower model greedy confidences at the end of training (Figure 1 (d)), consistent with Lemma 4.2.

We next assess pass@N test performance as a function of $N$ for models trained by minimizing $\mathcal{L}_{\text{DCO}}^{N'}$ for different values of $N'$ (Figure 2 (a)). For any given $N$, there is an optimal $N'$ for the training loss $\mathcal{L}_{\text{DCO}}^{N'}$ that maximizes pass@N test coverage, yielding a Pareto optimal performance frontier (black curve) that is achieved when $N'$ is close to $N$. In particular the model trained with CE loss (equivalent to pass@1 maximization at training) performs poorly relative to the Pareto frontier at large N (red curve below black at large $N$). Conversely, models trained with DCO at large $N'$ perform poorly relative to the Pareto frontier for small $N$ (green curves below black at small $N$).

Together these results indicate that the alignment of the training loss $\mathcal{L}_{\text{DCO}}^{N'}$ with the test time strategy pass@N, with $N'$ close to $N$, is crucial for obtaining Pareto optimal performance. Moreover, these results once again confirm the tradeoff between exploration and exploitation, with good performance using pass@N test-time strategies requiring high (low) exploration with low (high) confidence at large (small) $N$.

## 5.3 Improved theorem proving via ensembled tree search through a modified step-wise DCO

To further test our method in realistic settings with a verifier, we conduct experiments in theorem proving using an interactive proof assistant on the LeanDojo benchmark [44] extracted from the math

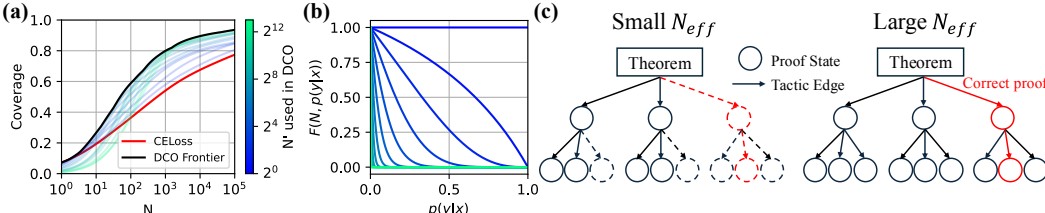

Figure 2: **(a) DCO improves on CE loss for pass@N test coverage over a broad range of N and traces a Pareto-optimal frontier.** We fine-tune Llama-3-8B base models on MATH to produce direct answers: one with CE loss and others using $\mathcal{L}_{\mathrm{DCO}}^{N'}$ for various $N'$ (color-coded). Each curve shows pass@N coverage for a *single* fine-tuned model. Note that no $N'$ is optimal for all $N$. The black curve is a Pareto-optimal performance frontier traced by the max of coverage curves for DCO over all $N'$. (See Figure 6, which highlights $N' = 256$, and Figures 8 and 10 for Llama-3-70B and AIME24 results.) **(b) DCO limits overconfidence by attenuating gradients for examples on which the model is highly confident.** We plot the confidence regularization factor $F$ in Equation (4). For CE loss ($N = 1$), $F = 1$ regardless of confidence $\hat{p}(y|x)$. For $N > 1$, $F$ decreases with $\hat{p}(y|x)$ and drops to zero around $1/N$—once confidence on an example reaches $O(1/N)$, $F$ vanishes, preventing it from further increasing. **(c) Inverse confidence $N_{\mathrm{eff}}$ at training time controls search-tree exploration shapes at test-time.** We provide schematic proof search trees for models trained with DCO$^{\mathrm{step}}$ for small (left) and large (right) $N_{\mathrm{eff}}$. Circles are proof states; edges are tactics; solid edges are explored, while dashed edges are not. Small $N_{\mathrm{eff}}$ yields a narrow search tree that may miss correct proofs (red, dashed), while larger $N_{\mathrm{eff}}$ expands the tree to include the correct proof. However, if the tree becomes too wide ($N_{\mathrm{eff}}^k \gg N$ for a $k$-step proof), sampling the correct path becomes unlikely and pass@N coverage decreases as $\mathcal{C}^N \approx N/N_{\mathrm{eff}}^k$. Thus $N_{\mathrm{eff}}$, chosen at training time, is a powerful knob to control the tradeoff between exploitation and exploration at test time.

library of LEAN4 [45]. In this task, at each step $i$ of the proof, the model is prompted with the current proof state $x[i]$, and it outputs a proof tactic $y[i]$ with a trainable model probability $\hat{p}(y[i]|x[i])$. The proof assistant takes the sampled tactic $y[i]$, verifies whether it is a valid tactic, and if so, returns the next proof state $x[i + 1]$ after the tactic $y[i]$ is applied. At test time this interactive process between the model and the assistant continues until either: (1) it terminates in an invalid tactic; (2) hits a maximal allowed search depth set by computational constraints; or (3) terminates in a successful proof of the initial proof goal as verified by the proof assistant.

In our experiments, we use LEAN4 [46] as the formal proof language. We start from Qwen2.5-Math-1.5B [28], and fine-tune it on the LeanDojo benchmark [44]. In contrast to solving math problems above where the model *first* autoregressively generates an entire answer $y$ according to $\hat{p}(y|x)$ and *then* the entire $y$ is verified for correctness, in theorem proving we *must* obtain correct tactics $y[i]$, as verified by the proof assistant, at *every* proof step $i$. A straightforward application of DCO fine-tuning in this setting then involves replacing the model confidence $\hat{p}(y|x)$ in Equation (3) on a single answer $y$, with the model confidence on an *entire* successful, complete $k$-step proof $\hat{p}(y[0], ..., y[k-1]|x[0]) = \prod_{i=0}^{k-1} \hat{p}(y[i]|x[i])$. Because of the chain rule, the DCO gradient on a single proof has the same form as Equation (4) but with $F(N, \hat{p}(y|x))$ replaced with $F(N, \prod_{i=0}^{k-1} \hat{p}(y[i]|x[i]))$. We naively applied this DCO fine-tuning method with $N' = 4k$ and evaluated the resulting model, as well as a base model trained with CE loss, on MiniF2F, using pass@4k. CE loss achieves a proof success rate of 37.4%, while fine-tuning with DCO achieves a 38.7% success rate. Thus, a naive application of DCO to theorem proving by matching the parameter $N'$ in DCO to the parameter $N$ in pass@N achieves only a modest improvement.

In theorem proving, since *every single* intermediate proof step tactic $y[i]$ must be valid, it is natural to consider a *step-wise generalization* of DCO. For example, at each step $i$, the model chooses a tactic $y[i]$ with confidence $\hat{p}(y[i]|x[i])$. Our proposed step-wise DCO optimizes this single-step confidence according to Equation (4) as before, except now the *step-wise* confidence regularizer is given by $F(N_{\mathrm{eff}}, \hat{p}(y[i]|x[i]))$. Here one can think of $N_{\mathrm{eff}}$ as a step-wise DCO hyperparameter that controls the exploration width at every proof step. In essence, the confidence regularizer prevents the confidence of any chosen tactic from becoming much higher than $1/N_{\mathrm{eff}}$. Since the sum of the confidences over all tactics at each step must be 1, this means that at test time, model exploration at each step corresponds to searching on a search tree in which each proof state $x[i]$ allows the exploration of

Table 2: Proof success rates on Mathlib and MiniF2F using pass@4k. DCO$^{\text{step}}$ outperforms CE loss ($N_{\text{eff}} = 1$) on both datasets for well-chosen $N_{\text{eff}}$. Ensembling models trained with different $N_{\text{eff}}$ leverages complementary strengths, with substantial gains over CE at matched test-time compute.

| $N_{\text{eff}}$ | 1 (CE loss) | 4 | 8 | 16 | 32 | Ensemble of all $N_{\text{eff}}$'s | $N_{\text{eff}} = 1$ (5x test compute) |
|---|---|---|---|---|---|---|---|
| Mathlib | 55.6% | 56.4% | 56.1% | **56.5%** | 55.8% | **62.2%** | 57.0% |
| MiniF2F | 37.4% | 39.0% | **39.5%** | 37.0% | 37.4% | **43.6%** | 39.5% |

approximately only $N_{\text{eff}}$ tactics. Thus using $N_{\text{eff}}$ in step-wise DCO during fine-tuning selects the approximate branching factor $N_{\text{eff}}$ of the proof search tree at test time (see Figure 2 (c)).

In particular, a valid (partial) proof of length $k$ will have probability of order $N_{\text{eff}}^{-k}$. Thus small $N_{\text{eff}}$ limits the exploration at test time to a tree with small branching factor, but allows longer proofs with larger numbers of steps $k$ to have higher sampling probability. This limited width search strategy should work well at test time using pass@N as long as two conditions hold: (1) a successful proof of length $k$ is present in the search tree of branching factor $N_{\text{eff}}$; and (2) the $N$ in pass@N at test time is large enough that this proof of probability $O(N_{\text{eff}}^{-k})$ is selected with high probability in $N$ passes (which starts to occur as soon as $N \gtrsim N_{\text{eff}}^k$). Conversely, larger $N_{\text{eff}}$ allows for more exploration at test time using a wider search tree of larger branching factor, but it makes finding longer proofs with large $k$ harder, since the approximate success condition for pass@N of $N > N_{\text{eff}}^k$ is harder to satisfy at fixed $N$ and large $N_{\text{eff}}$ and $k$. However, this wide exploration strategy can work well for short proofs with small $k$ if a successful short proof does not lie in the limited width search tree obtained at small $N_{\text{eff}}$, but does lie in a wider search tree at larger $N_{\text{eff}}$ (Figure 2, (c) right).

The mean and median lengths of the proofs in the training set are 4.2 and 2, respectively. Thus we explore fine-tuning with the step-wise DCO algorithm, DCO$^{\text{step}}$, with $N_{\text{eff}}$ ranging from 1 (CE loss) to $N_{\text{eff}} = 32$. We evaluate pass@4k performance on the Mathlib and MiniF2F test sets and find that larger $N_{\text{eff}}$ improves accuracy over the CE baseline (Table 2). Interestingly, the optimal $N_{\text{eff}}$ increases with more passes at test time (Table 6), similar to the Pareto frontier in Figure 2 (a). Since different choices of $N_{\text{eff}}$ correspond to different search strategies at test time—larger $N_{\text{eff}}$ favors short proofs in wide trees, smaller $N_{\text{eff}}$ favors long proofs in narrow trees— we hypothesized that ensembling these methods could significantly boost performance. This was indeed the case (ensemble entry in Table 2). The ensemble strategy samples $5 \times 4k$ times, so a proper baseline is CE loss with the same test-time compute of pass@N with $N = 20k$. The ensemble strategy outperforms this stringent baseline with significant excesses of 5.2% on Mathlib and 4.1% on MiniF2F.

These results indicate that varying $N_{\text{eff}}$ in DCO$^{\text{step}}$ at training time allows a diversity of tree search strategies trading depth and breadth that can be exploited by scaling test-time compute via pass@N.

### 5.4 Approximate DCO also improves performance with chain-of-thought reasoning

In Section 5.2, the model is trained to directly give an answer $y$ to a problem $x$. In this case, one can implement DCO at training time by explicitly computing the model confidence $\hat{p}(y|x)$ and using it in Equation (4). However, in a Chain-of-Thought (CoT) training paradigm, the training data consists of triplets $(x, c, y)$ where $x$ and $y$ are the problem and answer as before, but now $c$ is a CoT reasoning trace that explains how to derive $y$ from $x$. The model is trained on the triplet $(x, c, y)$, and at test time, given a new problem $x'$, it generates a reasoning trace $c'$ followed by an answer $y'$. Importantly, the model is evaluated on whether its answer $y'$ is correct *independent* of its reasoning trace $c'$.

Thus, to estimate the probability $\hat{p}(y|x)$ that the model assigns to any answer $y$, one can no longer compute it directly as in the direct answer case. Instead, one must marginalize over all possible reasoning traces $c$ to obtain $\hat{p}(y|x) = \sum_c \hat{p}(y, c|x)$. Exact marginalization is intractable, so we replace the marginalization with a Monte Carlo estimate of $\hat{p}(y|x)$, which we insert into the overconfidence regularization factor $F$ in Equation (4). We call this algorithm approximate DCO, denoted by DCO$^{\text{a}}$. We also compute Monte Carlo estimates of the probability the model assigns to its most likely final answer, $\hat{p}(y^{\text{mode}}|x)$, as a proxy for the model's max confidence. Note that in the CoT setting the greedy answer would correspond to a single *most* likely reasoning trace, but we want the probability of the most likely *final* answer marginalizing over *all* reasoning traces.

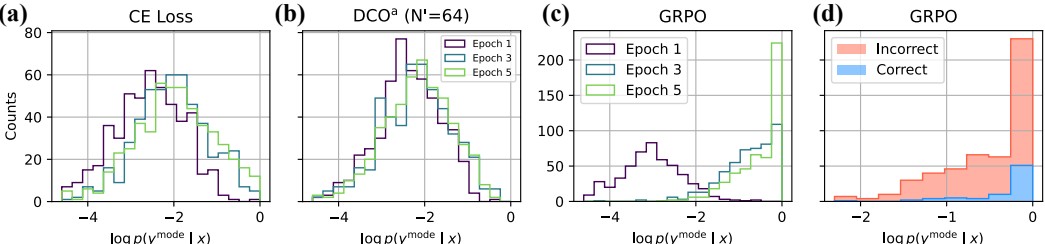

Figure 3: **(a, b) Overconfidence persists in CoT fine-tuning with CE loss.** We fine-tune a Llama-3-8B base model on CoT traces in the MATH dataset and plot the distribution of the estimated model confidences $\hat{p}(y^{\mathrm{mode}}|x)$ over samples in the test set at various points in training. **(a)** For CE loss, the model becomes more confident in its most likely answers as training progresses—the confidence distribution shifts to the right. This effect is milder than in the direct answer setting (Figure 1). **(b)** $\mathrm{DCO^a}$ limits this shift of the confidence distribution over training. **(c, d) Overconfidence is also present in GRPO fine-tuning and can result in a model that is overconfident *and* wrong. (c)** The model trained with GRPO assigns progressively larger confidences to $\hat{p}(y^{\mathrm{mode}}|x)$ over the course of training. **(d)** At the end of training, only a small portion of the model's highly confident completions are correct, which will harm the model's pass@N performance when scaling up $N$.

Table 3: Pass@N coverage on MATH ($\pm$ indicates standard error of the mean). We fine-tune Llama-3-8B-base on CoT traces. Training with $\mathrm{DCO^a}$ using $N' = 64$ underperforms CE loss for pass@1, but outperforms it for higher $N$, with the largest improvement when $N' = N$. The GRPO model has the strongest pass@1 performance, but its performance degrades at larger $N$ due to overconfidence.

| Pass@N | Epoch 1 | | | Epoch 3 | | | Epoch 5 | | |
|---|---|---|---|---|---|---|---|---|---|
| | CE | GRPO | $\mathrm{DCO^a}$ | CE | GRPO | $\mathrm{DCO^a}$ | CE | GRPO | $\mathrm{DCO^a}$ |
| Pass@1 | $4.3^{\pm0.1}$ | $3.0^{\pm0.1}$ | $\mathbf{5.0^{\pm0.1}}$ | $9.0^{\pm0.1}$ | $\mathbf{12.6^{\pm0.6}}$ | $8.3^{\pm0.1}$ | $9.9^{\pm0.1}$ | $\mathbf{14.6^{\pm0.1}}$ | $7.8^{\pm0.1}$ |
| Pass@16 | $30.2^{\pm0.7}$ | $27.1^{\pm0.7}$ | $\mathbf{34.2^{\pm0.6}}$ | $41.6^{\pm0.4}$ | $35.8^{\pm0.3}$ | $\mathbf{42.6^{\pm0.6}}$ | $40.9^{\pm0.4}$ | $29.2^{\pm0.2}$ | $\mathbf{42.8^{\pm0.5}}$ |
| Pass@64 | $51.2^{\pm1.2}$ | $50.6^{\pm0.7}$ | $\mathbf{55.3^{\pm1.0}}$ | $61.7^{\pm0.4}$ | $46.3^{\pm0.4}$ | $\mathbf{63.1^{\pm0.8}}$ | $60.9^{\pm0.6}$ | $35.4^{\pm0.3}$ | $\mathbf{64.3^{\pm0.6}}$ |

We conduct experiments on the MATH dataset. As a baseline, we first fine-tune a Llama-3-8B-base model using CE loss on the golden CoT solutions. We find in Table 3 that pass@1 coverage robustly improves over the course of CE loss training. Intriguingly, in the CoT setting, overfitting induced by the misalignment of CE loss at training time with the pass@N strategy at test time, appears less severe compare with the direct answer setting (cf. Table 1). Nevertheless, we still find a clear increase in the model's max confidence through training (Figure 3 (a)). In contrast, training with $\mathrm{DCO^a}$ mitigates the rise in the model's max confidence (Figure 3 (b)). As with the direct answer setup, model trained with $\mathrm{DCO^a}$ ($N' = 64$) exhibits lower pass@1 performance compared to the CE baseline but demonstrates superior performance when evaluated with pass@N for larger $N$ values (Table 3).

To provide a comprehensive comparison, we also evaluate a model trained with reinforcement learning. Specifically, we use GRPO [29] to train the base model with the r1 [10] prompt template. Interestingly, we observe that the GRPO-trained model exhibits pronounced overconfidence (Figure 3 (c)), yet only a small portion of the model's highly confident answers are correct (Figure 3 (d)). This negatively impacts the model's pass@N performance when scaling up $N$ (Table 3). Although GRPO demonstrates superior pass@1 performance, the model trained with $\mathrm{DCO^a}$ achieves a significantly stronger performance when evaluated with pass@N for larger $N$ values (Table 3).

Overall these results once again validate our theory and algorithmic approaches, now in a CoT reasoning setting. In essence, improved performance by scaling test time compute via a pass@N strategy at large $N$ can be best obtained by aligning the training strategy via choosing $\mathrm{DCO^a}$ with the same $N$, and the origin of the performance improvement comes from limiting model overconfidence.

## 6 Discussion

In summary, all our results suggest the need for a tight co-design of two traditionally separate phases of LLM development: (1) model training or fine-tuning and (2) test-time search/reasoning strategies

and budget. If the former is misaligned with the latter, then more training can actually *impair* performance gains from scaling test-time compute. But if they are properly co-designed, end-to-end training and test time performance can be far more effective. We have shown how to modify standard cross-entropy loss for training to be better aligned to a pass@N strategy for large $N$ at test-time. Moreover, we have suggested and empirically confirmed why this co-design of training loss and pass@N strategy is essential, because optimal policies for pass@N at large $N$ should unconfidently explore while the optimal policies for pass@N at small $N$ should confidently exploit.

**Limitations.** Our study primarily investigates SFT with the pass@N test-time strategy for its simplicity. However, this strategy requires a verifier, restricting its applicability to problems where verification is feasible. Extending our findings to other test-time strategies, such as best-of-N, is an important direction for future research. Additionally, while we have noted similar overconfidence phenomena in GRPO, a more comprehensive study of RL methods is beyond the current scope. Lastly, our work only focuses on mathematical reasoning; generalizing our findings to other domains, such as coding tasks where unit tests serve as natural verifiers, is left for future work.

## Acknowledgments and Disclosure of Funding

We would like to thank Zhinan Cheng, Clémentine Dominé, Marco Fumero, David Klindt, Daniel Kunin, Ben Sorscher and Atsushi Yamamura for helpful discussions. S.G. thanks the James S. McDonnell Foundation, Simons Foundation, NTT Research, Schmidt Foundation and an NSF CAREER Award for support. S.D. and F.C. were partially supported by the McKnight Foundation, the Simons Foundation and an NIH CRCNS grant R01DC020874. We are also grateful to Adam Ousherovitch and Hongda Yuan for identifying an error in Lemma 4.2, which has been corrected in the revised version of this paper.

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

# A  Proofs

## A.1  Proof of approximately well calibration under optimal policy

**Lemma A.1.** *Given any problem, let $\hat{p}_i$ denote the model probability or confidence assigned to answer $i$, and assume answers are sorted from highest to lowest confidence so that $\hat{p}_i \geq \hat{p}_{i+1}, \forall i \geq 1$. Let $p_i$ be the probability that answer $i$ is actually correct across all the problems. Then optimal policies maximizing pass@N coverage in Equation (2) (denoted by $\hat{p}_i^*$) is approximately well calibrated, i. e. higher confidence implies higher or equal probability of being correct, i.e. $p_i \geq p_{i+1}, \forall i \geq 1$.*

*Proof.* Let $R$ be the total number of answers the model can generate. Here $R$ could possibly equal infinity. Let $\hat{p}_i^*$ denote the model probability or confidence assigned to answer $i$ under optimal policy maximizing pass@N coverage in Equation (2). Let $A_i$ be the event that the $i^{\text{th}}$ ranked answer is not chosen in the $N$ passes and $B_i$ be the event that $i^{\text{th}}$ ranked answer is chosen in the $N$ passes. Let $1_{B_i}$ be the indicator function of $B_i$. Then, given an event $\omega$, the probability of getting the correct answer is $\sum_{i=1}^{R} p_i 1_{B_i}(\omega)$. Hence, the expected probability of getting the correct answer is

$$\mathbb{E}\left(\sum_{i=1}^{R} p_i 1_{B_i}\right) = \sum_{i=1}^{R} p_i \mathbb{P}(B_i) = \sum_{i=1}^{R} p_i(1 - \mathbb{P}(A_i)) = 1 - \sum_{i=1}^{R} p_i \mathbb{P}(A_i) = 1 - \sum_{i=1}^{R} p_i(1 - \hat{p}_i^*)^N, \quad (5)$$

If the current strategy is already optimal, any small change of $(\hat{p}_1^*, ..., \hat{p}_R^*)$ would not increase the expected probability of getting the correct answer. Now suppose we don't always have $p_i \geq p_{i+1}$, in other words, $p_j < p_{j+1}$ for some $j \geq 1$. If $\hat{p}_j^* < \hat{p}_{j+1}^*$, we have

$$\mathbb{E}\left(\sum_{i=1}^{R} p_i 1_{B_i}\right) < 1 - \sum_{i=1}^{j-1} p_i(1 - \hat{p}_i^*)^N - \sum_{i=j+1}^{R} p_i(1 - \hat{p}_i^*)^N - p_{j+1}(1 - \hat{p}_j^*)^N - p_j(1 - \hat{p}_{j+1}^*)^N, \quad (6)$$

where the last inequality says that we can increase the expected probability of getting the correct answer by swapping the confidence on answer originally labeled as $j$ and $j + 1$, which is a contradiction. If $\hat{p}_j^* = \hat{p}_{j+1}^*$, then for all $\delta > 0$ sufficiently small, we always have

$$\mathbb{E}\left(\sum_{i=1}^{R} p_i 1_{B_i}\right) < 1 - \sum_{i=1}^{j-1} p_i(1 - \hat{p}_i^*)^N - \sum_{i=j+1}^{R} p_i(1 - \hat{p}_i^*)^N - p_j(1 - \hat{p}_j^* - \delta)^N - p_{j+1}(1 - \hat{p}_{j+1}^* + \delta)^N,$$

$$(7)$$

which also contradict with the current policy being optimal. □

## A.2  Proof of Lemma 4.2

Assuming $p_1$ is known, an upper bound on $\hat{p}_1^*$ is given by,

$$u = \max_{\{p_s\}_{s=1}^{k}} \pi_1 \left( \operatorname*{argmin}_{\{\hat{p}_i\}_{i=1}^{k}} \left\{ \sum_{i=1}^{k} p_i(1 - \hat{p}_i)^N \right\} \right)$$

$$\text{s.t. } \sum_{s=1}^{k} p_s = 1 - \epsilon, \ p_1 \geq p_2 ... \geq p_k > 0. \quad (8)$$

Suppose $\{p_i\}_{i=1}^{k}$ solves Equation (8), and let $\{\hat{p}_i^*\}_{i=1}^{k}$ be the optimizer of

$$\operatorname*{argmin}_{\{\hat{p}_i\}_{i=1}^{k}} \{\sum_{i=1}^{k} p_i(1 - \hat{p}_i)^N\} \text{ s.t. } \sum_{s=1}^{k} \hat{p}_s = 1, \ 1 \geq \hat{p}_1 \geq 0, ..., 1 \geq \hat{p}_k \geq 0. \quad (9)$$

By definition, $u = \hat{p}_1^*$. Since $p_1 \geq p_2 ... \geq p_k$, we also have $\hat{p}_1^* \geq \hat{p}_2^* ... \geq \hat{p}_k^*$. There are $k$ possible cases.

1)  $\hat{p}_1^* = 1$ and $\hat{p}_s^* = 0$ for all $s \geq 2$.
j)  $\hat{p}_j^* > 0$ but $\hat{p}_{j+1}^* = 0$. For $2 \leq j \leq k - 1$. There are $k - 2$ cases.
k)  $\hat{p}_k^* > 0$.

We now characterize when each case can occur and derive an upper bound in each case.

### A.2.1 Case $k$

**Lemma A.2.** *If $\hat{p}_k^* > 0$, then $p_1 = p_2 = ... = p_{k-1}$.*

*Proof.* In this case, we have

$$\hat{p}_k^* = 1 - \frac{(k-1)p_k^{-\alpha}}{\sum_{s=1}^{k} p_s^{-\alpha}} > 0, \ \hat{p}_1^* = 1 - \frac{(k-1)p_1^{-\alpha}}{\sum_{s=1}^{k} p_s^{-\alpha}}. \tag{10}$$

where $\alpha = \frac{1}{N-1}$. Suppose, for contradiction, that $p_1 > p_{k-1}$. We construct a perturbed configuration $\{q_i\}_{i=1}^{k}$ such that the corresponding optimizer $\{\hat{q}_i^*\}_{i=1}^{k}$ satisfies $\hat{q}_1^* > \hat{p}_1^*$. Choose a small $\delta > 0$ and define

$$q_1 = p_1, \qquad \cdots \qquad , q_{k-2} = p_{k-2}, \qquad q_{k-1} = p_{k-1} + \delta, \qquad q_k = p_k - \delta.$$

For $\delta > 0$ sufficiently small, we still have $\hat{q}_k^* > 0$ (by continuity), and moreover $\hat{q}_1^* > \hat{p}_1^*$, contradicting the optimality of the original configuration $\{p_i\}_{i=1}^{k}$. Hence, $p_1 = p_2 = \cdots = p_{k-1}$. $\qquad\square$

By Lemma A.2, we have

$$(k-1)p_1 + p_k = 1 - \epsilon, \ (k-1)p_1^{-\alpha} > (k-2)p_k^{-\alpha}, \ p_1 \geq p_k > 0. \tag{11}$$

This implies the following range for $p_1$ in Case $k$:

$$\frac{1-\epsilon}{k} \leq p_1 < \frac{1-\epsilon}{k-1 + \left(\frac{k-2}{k-1}\right)^{N-1}}. \tag{12}$$

Under condition Equation (12), we have

$$u = 1 - \frac{(k-1)p_1^{-\alpha}}{(k-1)p_1^{-\alpha} + (1 + p_1 - \epsilon - kp_1)^{-\alpha}}. \tag{13}$$

Moreover, the right hand side of Equation (13) is an increasing function of $p_1$.

### A.2.2 Case $j$

**Lemma A.3.** *If $\hat{p}_j^* > 0$ but $\hat{p}_{j+1}^* = 0$ for some $2 \leq j \leq k-1$, then $p_1 = ... = p_{j-1}$ and $p_{j+1} = ... = p_k > 0$.*

*Proof.* We proceed in three steps.

**Step 1: we prove $p_k > 0$.** Suppose, by contradiction, $p_k = 0$. Let $s > 2$ be the smallest index such that $p_s = \cdots = p_k = 0$. If $s > j$, change $p_j$ to $p_j - \delta$, $p_{j+1}$ to $\delta$ for some small $\delta > 0$ If $s \leq j$, change $p_{s-1}$ to $p_{s-1} - \delta$, $p_j$ to $\delta$ for some small $\delta > 0$. In either case, for $\delta$ sufficiently small, the new system remains in Case $j$ but has strictly larger $\hat{p}_1^*$, a contradiction.

**Step 2: we prove $p_{j+1} = \cdots = p_k$.** Being in Case $j$ implies

$$
\begin{aligned}
&c_j > 0, \\
&c_r \leq 0, \ \forall j+1 \leq r \leq k, \\
&\text{where } c_r \equiv 1 - \frac{(r-1)p_r^{-\alpha}}{\sum_{t=1}^{r} p_t^{-\alpha}}.
\end{aligned} \tag{14}
$$

If $c_{j+1} < 0$, change $p_j$ to $p_j - \delta$, $p_{j+1}$ to $p_{j+1} + \delta$ for some small $\delta > 0$. The new system is still in Case $j$ but with strictly larger $\hat{p}_1^*$, which leads to contradiction. Hence $c_{j+1} = 0$.

If $c_{j+2} < 0$, then $p_{j+2} < p_{j+1}$ and we can change $p_{j+1}$ to $p_{j+1} - \delta$, $p_{j+2}$ to $p_{j+2} + \delta$ for some small $\delta > 0$. Under this change, $c_{j+1}$ decreases, $c_{j+2}$ increases, $c_s, s \geq j+3$ decreases, so the configuration remains in Case $j$ but with equal $\hat{p}_1^*$ for $\delta > 0$ sufficiently enough. But then $c_{j+1} < 0$, contradicting $c_{j+1} = 0$. Iterating this argument yields $c_{j+1} = \cdots = c_k = 0$, which implies $p_{j+1} = \cdots = p_k$.

**Step 3: We prove $p_1 = \cdots = p_{j-1}$.** This follows from the same argument as in the proof of Lemma A.2. $\qquad\square$

As a direct consequence, we obtain the following corollary.

**Corollary A.4.** *Let $p_j = q$, $p_{j+1} = ... = p_k = p$. The upper bound $u$ is given by*

$$u = 1 - \frac{(j-1)p_1^{-\frac{1}{N-1}}}{(j-1)p_1^{-\frac{1}{N-1}} + q^{-\frac{1}{N-1}}} = \frac{q^{-\frac{1}{N-1}}}{(j-1)p_1^{-\frac{1}{N-1}} + q^{-\frac{1}{N-1}}}. \tag{15}$$

*Moreover,*

$$\frac{1}{j} \leq u < \frac{1}{j-1}. \tag{16}$$

*and $\{q, p, j\}$ satisfies*

$$\begin{aligned}
&p_1 \geq q \geq p > 0, \\
&(j-1)p_1 + q + (k-j)p = 1 - \epsilon, \\
&(j-1)(p^{-\alpha} - p_1^{-\alpha}) = q^{-\alpha}, \ \alpha = \frac{1}{N-1}, \\
&q > \left(\frac{j-2}{j-1}\right)^{N-1} p_1.
\end{aligned} \tag{17}$$

*Proof.* This is an immediate consequence of Lemma A.3 and its proof (using $c_{j+1} = 0$, $c_j > 0$). □

Once $j$ is determined, we can compute $q$ and hence $u$. Equation (17) determines $j$ in the following sense.

**Lemma A.5.** *If $\{q_0, p_0, j_0\}$ satisfies Equation (17), then $j_0 \geq j$.*

*Proof.* Let $u_0$ be the corresponding maximal value associated with $\{q_0, p_0, j_0\}$. Then $u_0 \leq u$. Combining Equation (17) and Equation (15) we have

$$\frac{1}{j_0} \leq u_0 < \frac{1}{j_0 - 1}. \tag{18}$$

If $j < j_0$, then $u_0 > u$, which is a contradiction. □

**Lemma A.6.** *For any fixed $\epsilon$, $p_1$ and $k$, $j = j(N)$ is a non-decreasing function of $N$. Since $1/j \geq u > 1/(j-1)$, $u = u(p_1)$ is a non-decreasing function of $N$.*

*Proof.* Suppose, for contradiction, that $j(N+1) < j(N)$. Then

$$\left(\frac{j(N+1)-2}{j(N+1)-1}\right)^N < \left(\frac{j(N)-2}{j(N)-1}\right)^{N-1}. \tag{19}$$

Let $\{j(N+1), q(N+1), p(N+1)\}$ be the corresponding configuration at $N+1$. Using the inequality above, this configuration can be used to construct a larger value of $u$ at $N$ with index $j = j(N+1) < j(N)$, contradicting the minimality of $j$ in Lemma A.5. □

### A.2.3 Case 1

This case is impossible.

### A.2.4 Summary of the Proof of Lemma 4.2

- If

$$\frac{1-\epsilon}{k} \leq p_1 < \frac{1-\epsilon}{k-1 + \left(\frac{k-2}{k-1}\right)^{N-1}}, \tag{20}$$

then Case $k$ applies. In this regime, the upper bound $u$ is an increasing function of $p_1$ and is given by

$$u = 1 - \frac{(k-1)p_1^{-\frac{1}{N-1}}}{(k-1)p_1^{-\frac{1}{N-1}} + (1 + p_1 - \epsilon - kp_1)^{-\frac{1}{N-1}}}. \tag{21}$$

- Otherwise, Case $j$ applies, and the upper bound is

$$u = 1 - \frac{(j-1)p_1^{-\frac{1}{N-1}}}{(j-1)p_1^{-\frac{1}{N-1}} + q^{-\frac{1}{N-1}}}. \tag{22}$$

For any fixed $\epsilon$, $p_1$ and $k$, let $j$ be the smallest positive integer such that the following equations (viewing $\{q, p\}$ as variables) admits a solution:

$$\begin{aligned}
& p_1 \geq q \geq p > 0, \\
& (j-1)p_1 + q + (k-j)p = 1 - \epsilon, \\
& (j-1)(p^{-\alpha} - p_1^{-\alpha}) = q^{-\alpha}, \ \alpha = \frac{1}{N-1}. \\
& q > \left(\frac{j-2}{j-1}\right)^{N-1} p_1.
\end{aligned} \tag{23}$$

The index $j$ is non-decreasing with respect to $N$, and $u$ is further bounded by

$$\frac{1}{j} \leq u < \frac{1}{j-1}. \tag{24}$$

### A.3 Proof of Lemma 4.3

*Proof.* The following proof is motivated by the following intuition: given the values of $p_1$ and $p_2$, $\hat{p}_1^*$ attains its smallest possible value when the $p_j$ ($j \geq 3$) are as large as possible. Let $s$ be the smallest integer such that $p_1 + sp_2 \geq 1$. We have,

$$\hat{p}_1^* = \pi_1 \left( \underset{\{\hat{p}_i\}_{i=1}^R}{\operatorname{argmin}} \{ \sum_{i=1}^R p_i(1 - \hat{p}_i)^N \} \right) \geq \pi_1 \left( \underset{\{\hat{p}_i\}_{i=1}^{s+1}}{\operatorname{argmin}} \{ p_1(1 - \hat{p}_1)^N + \sum_{i=2}^{s+1} p_2(1 - \hat{p}_i)^N \} \right). \tag{25}$$

Second, for a fixed $\hat{p}_1$, Jensen's inequality tells us that

$$p_1(1 - \hat{p}_1)^N + \sum_{i=2}^{s+1} p_2(1 - \hat{p}_i)^N \geq p_1(1 - \hat{p}_1)^N + sp_2(1 - \frac{1 - \hat{p}_1}{s})^N \tag{26}$$

Therefore, we have

$$\pi_1 \left( \underset{\{\hat{p}_i\}_{i=1}^{s+1}}{\operatorname{argmin}} \{ p_1(1 - \hat{p}_1)^N + \sum_{i=2}^{s+1} p_2(1 - \hat{p}_i)^N \} \right) = \underset{\hat{p}_1}{\operatorname{argmin}} \{ p_1(1 - \hat{p}_1)^N + sp_2(1 - \frac{1 - \hat{p}_1}{s})^N \} \tag{27}$$

Computing the right hand side of Equation (27) by taking derivatives we have

$$\hat{p}_1^* \geq \underset{\hat{p}_1}{\operatorname{argmin}} \{ p_1(1 - \hat{p}_1)^N + sp_2(1 - \frac{1 - \hat{p}_1}{s})^N \} = 1 - \frac{sp_1^{\frac{1}{1-N}} p_2^{\frac{1}{N-1}}}{s + p_1^{\frac{1}{1-N}} p_2^{\frac{1}{N-1}}}. \tag{28}$$

Since $s$ is the smallest integer such that $p_1 + sp_2 \geq 1$, we have

$$1 + p_2 > p_1 + sp_2, \tag{29}$$

Combining Equation (29) and Equation (28) we arrive at

$$\hat{p}_1^* \geq 1 - \frac{(1 - p_1 + p_2)p_1^{\frac{1}{1-N}} p_2^{\frac{1}{N-1}}}{1 - p_1 + p_2 + p_1^{\frac{1}{1-N}} p_2^{\frac{N}{N-1}}}. \tag{30}$$

$\square$

# B Experimental details

Our codebase uses PyTorch [47], Accelerate [48], and deepspeed (`https://github.com/microsoft/DeepSpeed`) to enable efficient training with memory constraints, and vllm [49] for efficient inference. Code will be made available here (https://github.com/allanraventos/refine).

**MATH.** For experiments with MATH dataset [43] (license: MIT), we fine-tune the Llama-3-8B-base [42] on the MATH [43] dataset. We start from the base model rather than Llama-3-8B-Instruct to avoid potential leakage of the MATH dataset into Llama-3-8B-Instruct through post-training process. We follow Lightman et al. [24] and use $12,000$ problems for training and the remaining $500$ for testing. In Sections 4 and 5.2, Figure 1, Figure 2 (a) and (b), we fine-tune the model for 4 epochs with a learning rate of 2e-5 and batch size 64. We adopt a linear learning rate warmup in the first 20 steps. For experiments with DCO, some of the data may have an extreme confidence regularizer value, if the model is already quite confident in answers to certain problems. To maintain an approximately fixed batch size, we set a threshold of $0.3$ on the confidence regularizer $F$. Any training data examples with an $F$ lower than this threshold will be replaced with new training examples to construct the batch. For the CoT experiments in Section 5.4, we use learning rate 2e-5 and batch size 128, with the same learning rate warmup.

**Theorem proving.** We use LeanDojo benchmark[44] (license: CC-BY 2.0) and adopt the random train and test split as introduced in Yang et al. [44]. The random test set includes 2,000 theorems. We fine-tune the model Qwen2.5-Math-1.5B [28] on the training set for 3 epochs with learning rate 1e-5 and batch size 64. We adopt a linear learning rate warmup in the first 20 steps. To evaluate the model, we use LeanDojo [44] to interact with the proof assistant. We impose a maximum wall clock time for each theorem, in addition to limiting the number of passes per problem. For experiments with 4k passes, the time budget is fixed at 5,000 seconds. In order to avoid the model going infinitely deep in the search tree, we limit the number of proof steps to be at most 50.

**DCO$^a$ objective.** The DCO$^a$ introduced in Section 5.4 is an approximation for DCO. To construct a batch of size $B$ with DCO$^a$, we process batches of samples sequentially; for each batch, we run online inference on each of the samples and discard all samples with probability of success rate larger than $p^{\text{thresh}}$. We choose to discard samples which have a DCO confidence reguarizer $F$ lower than $0.01$, corresponding to $p^{\text{thresh}} = 0.1$ for $N' = 64$. This process continues until we have enough training data for a single batch. In Figure 4, we plot the number of discarded samples as a function of training step for our DCO$^a$ experiments (same ones as in Table 3).

**GRPO.** We perform GRPO fine-tuning [10, 29] on problems of all difficulty levels in the MATH training set. We follow the modifications to GRPO proposed in Dr. GRPO ([33]), that is, we do not normalize advantages by the standard deviation of the group rewards, do not normalize a roll-out's contribution to the loss by its length, and do not include a KL divergence term with respect to the reference model. Furthermore, we perform only *one* policy gradient update for each set of model roll-outs, so there is no need for clipping. This leads to the surrogate objective taking the form,

$$\mathcal{L}(\theta) = \frac{1}{N'} \sum_{i=1}^{N'} \sum_{t=1}^{|o_i|} \frac{\pi_\theta(o_{i,t} \mid q, o_{i,<t})}{\text{stop\_grad}\left(\pi_\theta(o_{i,t} \mid q, o_{i,<t})\right)} \hat{A}_{i,t}$$

where we adopt notation similar to [33]; $N'$ is the number of roll-outs for a single problem statement $q$, $o_i$ is the $i$th roll-out in the group, $o_{i,t}$ is the $t$th token in $o_i$, and $\pi_\theta(o_{i,t} \mid q, o_{i,<t})$ is the probability assigned by the model to token $o_{i,t}$ conditioned on context $o_{i,<t}$. The $\text{stop\_grad}$ operator prevents the flow of gradients when computing $\nabla_\theta \mathcal{L}$. The advantages, in turn, are computed as $\hat{A}_{i,t} = r(q, o_i) - \frac{1}{N'} \sum_{i'=1}^{N'} r(q, o_{i'})$, where $r(q, o_i)$ is the binary reward assigned by the verifier to the solution $o_i$ for the problem $q$.

For our experiments, we use $N' = 8$ and batch size 128, which results in a total of 1,024 roll-outs per batch. We also perform batch balancing such that each batch contains no problems for which either $r(q, o_i) = 0$ for all $i$ or $r(q, o_i) = 1$ for all $i$, since these samples do not provide any gradient signal. We use a constant learning rate of 1e-6,with a 20-step linear warmup; we train for 5 epochs. Note that batch balancing results in fewer gradient steps. Additionally, we use the r1 template from [10] for our experiments,

```
A conversation between User and Assistant. The User asks a question, and the
  Assistant solves it. The Assistant first thinks about the reasoning process in
```

```
the mind and then provides the User with the answer. The reasoning process is
enclosed within <think> </think> and the answer is enclosed within <answer> </
answer> tags, respectively, i.e., <think> reasoning process here </think> <answer
> answer here </answer>.
User: {question}
Assistant: <think>
```

We strictly require solutions to follow the format specified in the prompt.

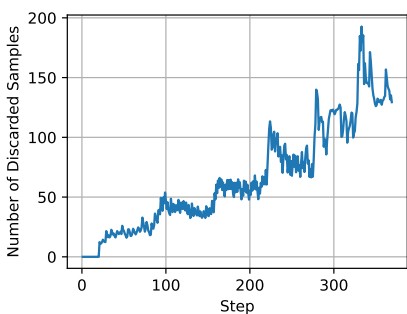

Figure 4: Number of discarded samples as a function of training step for the DCO[a] experiments. The
step structure reflects the model revisiting examples it has seen previously in training, where each
step closely matches the start of a new epoch.

**Implementing online inference.** Throughout our experiments and analysis, we extensively use the
open-source vllm package [49] for efficient inference. Integrating inference into the training loop
to enable training under the DCO[a] objective, as in Section 5.4, poses a challenging implementation
problem. We solve this problem by placing vllm worker processes, which together perform inference
on all GPUs, in a separate process group and use Ray (`https://github.com/ray-project/ray`)
to isolate them from the training loop. This enables running concurrent training and inference on the
same set of GPUs. We believe this inference in the loop setup will be useful to the community, as it
enables straightforward implementation of online data filtering approaches for LLM training.

**Computational cost.** All experiments with model size smaller than 10B are performed on machines
with 8 NVIDIA H100 GPUs or 8 NVIDIA A100 GPUs. Experiments with model size larger than
10B are performed on machines with 8 NVIDIA H200 GPUs or 8 NVIDIA B200 GPUs. Specifically,
for DCO on the MATH dataset, each run required approximately 1 NVIDIA H100 GPU-hour when
using the Llama-3-8B-base model, and approximately 4 NVIDIA B200 GPU-hours when using the
Llama-3-70B-base model. For DCO[step] on the LeanDojo benchmark with the Qwen2.5-Math-1.5B
model, each run required around 12 NVIDIA H100 GPU-hours. For DCO[a] on the MATH dataset
with CoT, each run consumed about 60 NVIDIA H100 GPU-hours for the Llama-3-8B-base model.

For both DCO and DCO[step], the additional computational cost is minimal—limited to computing the
DCO factor, which is negligible compared to the cost of forward and backward passes. However,
DCO[a], when applied to fine-tuning with CoT traces, introduces significant overhead due to the need
for online Monte Carlo estimation. In the setting of Llama-3 8B, with 32 layers, hidden dimension
4096, context length 1024, and batch size 128, we estimate 5 petaflops for each forward-backward
pass. Generating $N_{MC} = 64$ CoT's for each of 128 examples, in turn, requires approximately 50
petaflops. The actual inference compute cost depends on model performance: stronger models, which
will discard more samples, will need to process more samples in order to construct a full batch for the
next training step. Therefore there is a tradeoff: constructing a batch of 128 **hard** samples requires
running inference on more samples, but also means fewer steps are required to make a full pass
through the dataset.

The memory overhead is approximately 30% of GPU memory, when running one inference engine
per pair of H100 80GB GPUs. We note that while the flop overhead is significant, costly online
inference is also required in RL frameworks for CoT training. Furthermore, the choice of $N_M C$ can
be made smaller, leading to less precise estimates of the model's probability of success on a given
problem, while speeding up training.

# C   Additional results

## C.1   Easy data drives overconfidence

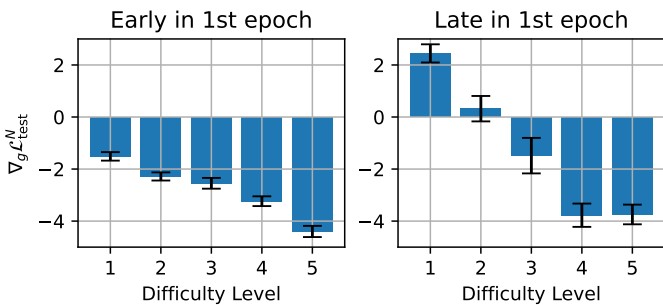

Figure 5: **Easy data drives overconfidence and degrades performance when scaling test-time compute**. For the experiments in Figure 1, we compute the directional derivative $\nabla_g \mathcal{L}_{\text{test}}^N$ at two points during the first training epoch: 11% (early) and 86% (late). At each stage, the gradient direction $g = -\sum_{(x_i, y_i) \in \mathcal{B}} \nabla \ell_{\text{CE}}(x_i, y_i)$ is evaluated on batches of **previously unseen** training data of each difficulty level defined in the MATH dataset. Early in training, data from all difficulty levels contribute to decreasing the test loss (*left*). However, later in training, easier examples (difficulty level 2) provide no further benefit, while the easiest examples (difficulty level 1) actively degrade test performance (*right*). The plotted $\nabla_g \mathcal{L}_{\text{test}}^N$ is an average over batches of unseen data, and we use $N = 256$, corresponding to pass@256.

We have shown in Section 4.2 that overconfidence limits performance gains from scaling test-time compute. Here we investigate whether all training data contribute equally to the drop in pass@N test performance. To quantify this effect, we define the test loss as the negative log-probability of coverage, $\mathcal{L}_{\text{test}}^N = -\sum_{(x,y) \in \mathcal{D}^{\text{test}}} \log \mathcal{C}_{(x,y)}^N$, where $\mathcal{C}_{(x,y)}^N$ denotes the coverage for a single test example $(x, y)$. To measure how a training batch $\mathcal{B}$ influences the final test loss, we compute the directional derivative $\nabla_g \mathcal{L}_{\text{test}}^N$ for the gradient direction $g = -\sum_{(x_i, y_i) \in \mathcal{B}} \nabla \ell_{\text{CE}}(x_i, y_i)$, where $\ell_{\text{CE}}$ is the standard CE loss on a single example. A negative value, $\nabla_g \mathcal{L}_{\text{test}}^N < 0$, indicates that a gradient step in the direction of $g$ decreases the test loss.

To analyze the impact of data difficulty, we group training data by difficulty level (as specified in the MATH dataset) and use the directional derivative to examine how a batch of **previously unseen** data from a **given difficulty level** would affect $\mathcal{L}_{\text{test}}^N$ at different stages of training. Early in training, we observe that $\nabla_g \mathcal{L}_{\text{test}}^N$ is negative across all difficulty levels, indicating that data from all difficulty levels contribute positively to reducing the test loss (Figure 5, left). However, near the end of the first epoch, training on easier examples (difficulty levels 1 and 2) no longer improves performance, and the easiest examples (level 1) actively degrade pass@N test performance (Figure 5, right). This shows how continued training on easy data can harm test performance when scaling test-time compute.

## C.2 Optimality of DCO is achieved when the training objective aligns with the test objective.

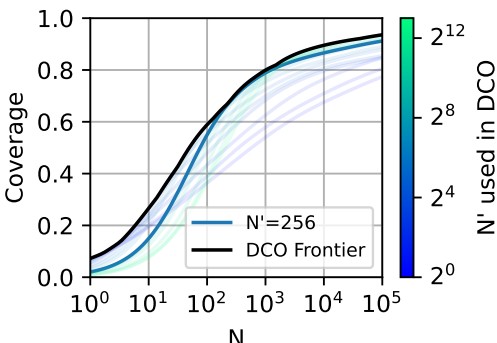

Figure 6: **Pareto-optimality at a given test strategy pass@N for some $N$ is obtained by DCO training for some $N'$ close to $N$.** Same as Figure 1 but with $N' = 256$ highlighted. We fine-tune Llama-3-8B base models on the MATH dataset to produce direct answers. We fine-tune for 4 epochs one model using CE loss and several models under the $\mathcal{L}_{\text{DCO}}^{N'}$ objective, for choices of $N'$ indicated by color. Note that there is no choice of $N'$ that is optimal across all $N$ (different colors are higher at different $N$). The black curve is a Pareto-optimal performance frontier traced by the max of coverage curves for DCO over all $N'$. Fine-tuning with the DCO loss ($N' = 256$) achieves Pareto-optimality for $180 \leq N \leq 385$ at test-time, further confirming that Pareto-optimality at a given test strategy pass@N for some $N$ on the x-axis is obtained by DCO training for some $N'$ close to $N$.

## C.3 Additional results with Llama-3-70B-base

In this section, we present additional results using the larger Llama-3-70B-base model, which are consistent with the findings reported for Llama-3-8B-base.

Table 4: Same as Table 1 but with Llama-3-70B-base model. Pass@N coverage metric on the MATH test set for a Llama-3-70B-base model fine-tuned with CE loss on direct answers from the MATH training set. Surprisingly, Pass@N test accuracy at large $N$ decreases with number of training epochs.

|         | pass@1   | pass@16   | pass@256  | pass@4k   |
|---------|----------|-----------|-----------|-----------|
| Epoch 1 | 5.9%     | **34.3%** | **68.8%** | **84.2%** |
| Epoch 2 | 6.9%     | 32.0%     | 63.0%     | 81.5%     |
| Epoch 3 | **8.2%** | 24.2%     | 48.8%     | 72.2%     |
| Epoch 4 | 7.9%     | 22.2%     | 44.2%     | 65.9%     |

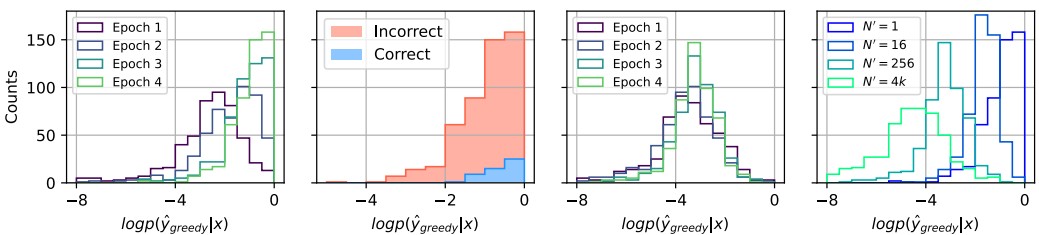

Figure 7: **Larger-sized model trained with CE loss also becomes overconfident in its greedy completions; our proposed DCO objective limits this overconfidence on larger-sized models.** Same as Figure 1 but with Llama-3-70B-base. We fine-tune a Llama-3-70B-base model on the MATH dataset to produce direct answers without a reasoning trace. $\hat{y}_{\text{greedy}}$ is the model's greedy completion when sampling autoregressively and choosing the most likely token at each step. **Leftmost:** The model trained with CE loss assigns progressively larger confidences $\hat{p}(\hat{y}_{\text{greedy}}|x)$ to its greedy completions over the course of training. **Left:** At the end of the training, only a small portion of the model's highly confident completions are correct. This will harm the model's pass@N performance when scaling up $N$. **Right:** Same as leftmost but shown for the DCO loss with $N = 256$. Relative to the CE loss, the model trained on DCO shows a much milder overconfidence effect. **Rightmost:** The confidence distribution of the greedy completions after four epochs with DCO for various choices of $N$. As $N$ increases, the model's confidence on the greedy completion is more stringently limited, directly as a consequence of the overconfidence regularizer $F$.

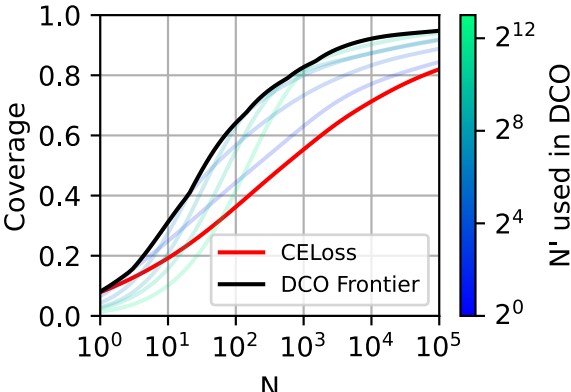

Figure 8: **DCO improves on CE for pass@N test coverage over a broad range of N and traces a Pareto-optimal frontier with Llama-3-70B-base.** Same as Figure 2 (a) but with Llama-3-70B-base model. We fine-tune Llama-3-70B base models on the MATH dataset to produce direct answers. We fine-tune for 4 epochs one model using CE loss and several models under the $\mathcal{L}_{\text{DCO}}^{N'}$ objective, for choices of $N'$ indicated by color. We plot pass@N test coverage as a function of $N$, with each curve (solid red or faint blue-green) corresponding to *one* fine-tuned model. Note that there is no choice of $N'$ that is optimal across all $N$ (different colors are higher at different $N$). The black curve is a Pareto-optimal performance frontier traced by the max of coverage curves for DCO over all $N'$. Pareto-optimality at a given test strategy pass@N for some $N$ on the x-axis is obtained by DCO training for some $N'$ close to $N$.

## C.4 Out-of-distribution evaluation

In this section, we present additional evaluation results on AIME24. We confirm that overconfidence persists even on the out-of-distribution test set, and that the DCO algorithm saves test-time scaling by regularizing the max confidence.

Table 5: Same as Table 1 but evaluate on AIME24 dataset. Pass@N coverage metric on the AIME24 dataset for a Llama-3-8B-base model fine-tuned with CE loss on direct answers from the MATH training set. Surprisingly, Pass@N test accuracy at large $N$ decreases with number of training epochs.

|         | pass@1 | pass@16 | pass@256 | pass@4k |
|---------|--------|---------|----------|---------|
| Epoch 1 | 0.2%   | 3.4%    | 30.5%    | **80.5%** |
| Epoch 2 | 0.3%   | 4.1%    | **31.7%** | 75.6%  |
| Epoch 3 | 0.2%   | 2.9%    | 16.0%    | 42.0%   |
| Epoch 4 | **1.9%** | **4.9%** | 15.2%   | 35.8%   |

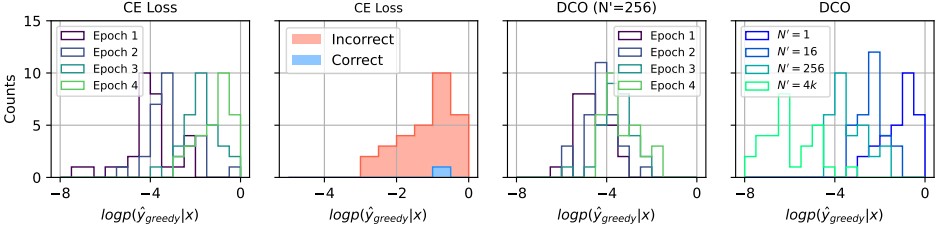

Figure 9: **A model trained with CE loss becomes overconfident in its greedy completions on out-of-distribution test set; our proposed DCO objective limits this overconfidence on out-of-distribution test set.** Same as Figure 1 but evaluated on AIME24. We fine-tune a Llama-3-8B-base model on the MATH dataset to produce direct answers without a reasoning trace and evaluate the model on AIME24. $\hat{y}_{\text{greedy}}$ is the model's greedy completion when sampling autoregressively and choosing the most likely token at each step. **Leftmost:** The model trained with CE loss assigns progressively larger confidences $\hat{p}(\hat{y}_{\text{greedy}}|x)$ to its greedy completions over the course of training. **Left:** At the end of the training, only a small portion of the model's highly confident completions are correct. This will harm the model's pass@N performance when scaling up $N$. **Right:** Same as leftmost but shown for the DCO loss with $N = 256$. Relative to the CE loss, the model trained on DCO shows a much milder overconfidence effect. **Rightmost:** The confidence distribution of the greedy completions after four epochs with DCO for various choices of $N$. As $N$ increases, the model's confidence on the greedy completion is more stringently limited, directly as a consequence of the overconfidence regularizer $F$.

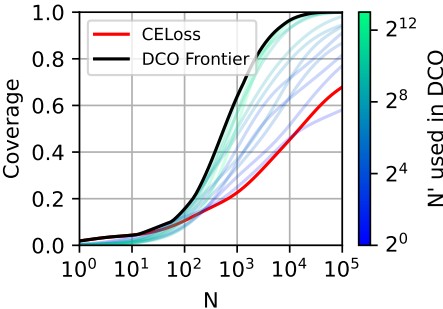

Figure 10: **DCO improves on CE for pass@N test coverage over a broad range of N and traces a Pareto-optimal frontier on out-of-distribution test set.** Same as Figure 2 (a) but evaluated on AIME24. We fine-tune Llama-3-8B base models on the MATH dataset to produce direct answers and evaluate on AIME24. We fine-tune for 4 epochs one model using CE loss and several models under the $\mathcal{L}_{\text{DCO}}^{N'}$ objective, for choices of $N'$ indicated by color. We plot pass@N test coverage as a function of $N$, with each curve (solid red or faint blue-green) corresponding to *one* fine-tuned model. Note that there is no choice of $N'$ that is optimal across all $N$ (different colors are higher at different $N$). The black curve is a Pareto-optimal performance frontier traced by the max of coverage curves for DCO over all $N'$. Pareto-optimality at a given test strategy pass@N for some $N$ on the x-axis is obtained by DCO training for some $N'$ close to $N$.

## C.5  Comparing with Focal Loss

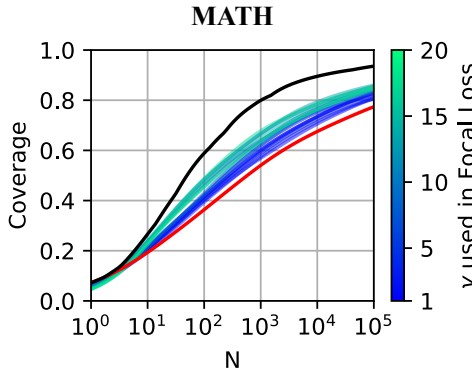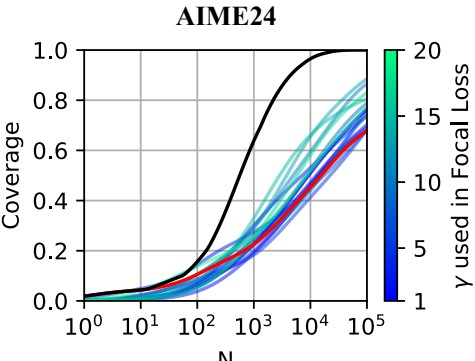

Figure 11: **DCO outperforms Focal Loss [50] for pass@N test coverage over a broad range of** $N$ **both for in- and out-of-distribution test sets.** We fine-tune Llama-3-8B base models on the MATH dataset to produce direct answers, using the Focal Loss [50]: $\mathcal{L}_{\text{FL}}^{\gamma}(x, y) = -(1 - \hat{p}(y \mid x))^{\gamma} \log \hat{p}(y \mid x)$. We fine-tune several models for 4 epochs under $\mathcal{L}_{\text{FL}}^{\gamma}$ for choices of $\gamma$ indicated by color. We plot pass@N test coverage as a function of $N$ for MATH (**left**) and AIME24 (**right**), with each curve corresponding to *one* fine-tuned model. The black curve shows the Pareto-optimal performance frontier traced by the max of coverage curves for DCO over all $N'$. The CE loss curves (red) and the DCO frontier curves (black) are the same as in Figure 2 (a) for MATH and Figure 10 for AIME24.

## C.6  Theorem proving

In Table 6, we show additional results for model performance under the DCO$^{\text{STEP}}$ objective. We find that the optimal $N_{\text{eff}}$ grows with increasing passes, agreeing with results in Sections 5.2 and 5.4. We also conduct expert iteration [51, 52] on Mathlib with theorems that do not have proof traces in the training set. We use pass@1k to prove those theorems. We find that our algorithm achieves a stronger improvement over the baseline for pass@4k after the 1$^{\text{st}}$ iteration. This improvement might result from the fact that the models can prove more easy theorems where the model has a higher confidence. As a result, we believe our method will perform better with expert iteration.

Table 6: Success rate on lean-dojo benchmark random test set trained with DCO$^{\text{step}}$.

| DCO$^{\text{STEP}}$ | EXPERT ITERATION 0 | | | | | EXPERT ITERATION 1 | | |
|---|---|---|---|---|---|---|---|---|
| | PASS@16 | PASS@64 | PASS@256 | PASS@1K | PASS@4K | PASS@16 | PASS@256 | PASS@4K |
| $N_{\text{eff}} = 1$ (CE) | **30.0%** | 38.75% | 46.05 | 50.75% | 55.55% | 40.3% | 52.65% | 58.55% |
| $N_{\text{eff}} = 4$ | **30.15%** | **39.5%** | **47.2%** | **52.95%** | 56.35% | **40.8%** | **53.05%** | 59.45% |
| $N_{\text{eff}} = 8$ | **30.2%** | 38.9% | **47.15%** | 52.7% | 56.1% | 40.1% | **53.25%** | 59.5% |
| $N_{\text{eff}} = 16$ | 28.65% | 46.7% | 46.45% | **52.9%** | **56.5%** | 39.05% | 52.8% | **60.05%** |
| $N_{\text{eff}} = 32$ | 26.05% | 46.7% | 45.6% | 51.5% | 55.8% | 37.05% | 52.2% | 59.15% |
| ENSEMBLE | 40.6% | 49.15% | 54.6% | 59.0% | 62.15% | 49.05% | 59.3% | 64.8% |

## C.7 Plot of the upper bound and the lower bound

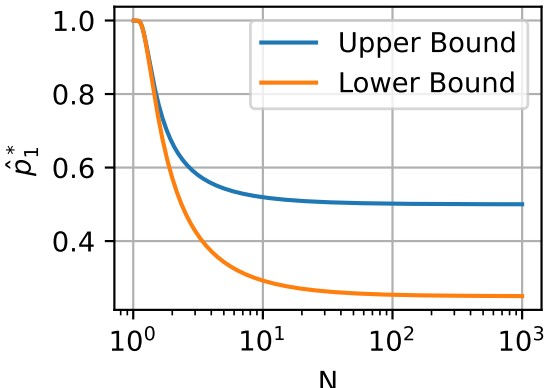

Figure 12: **Plot of the upper bound (Appendix A.2.4) and the lower bound (Equation (28)).** We plot the upper bound (blue) and lower bound (orange) for $p_1 = \frac{1}{2}$, $p_2 = \frac{1}{4}$, $\epsilon = \frac{1}{4}$ and $k = 2$. Both the upper bound and the lower decrease monotonically in $N$ and they both tends to 1 as $N \to 1^+$.

## C.8 Empirical evaluation of the approximately well calibrated assumption

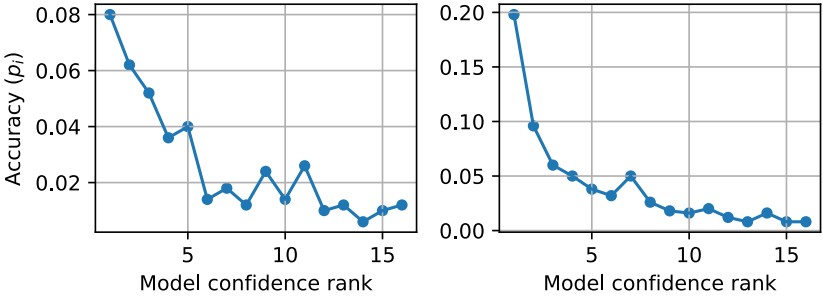

Figure 13: **We empirically verify the assumption that models are approximately well-calibrated.** **Left**: We fine-tune a Llama-3-8B-base model with CE loss on the MATH dataset **without** CoT and perform beam search with a width of 256 to obtain the top 16 most probable completions. We then measure the accuracy of these completions at each confidence rank. **Right**: We fine-tune a Llama-3-8B-base model with CE loss on the MATH dataset **with** CoT and perform estimate the top 16 most frequent answers after tracing out the reasoning traces. We then measure the accuracy of these answers at each frequency rank. The results demonstrate that test accuracy decreases approximately monotonically with model confidence rank, supporting the assumption.

**C.9    The data dependency of confidence and factor $F$**

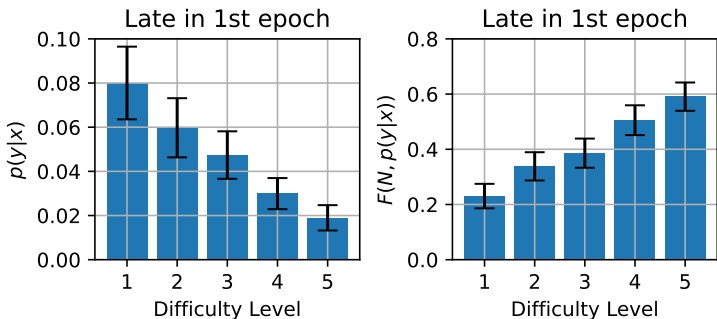

Figure 14: **Model is more confident on easy problems; DCO improves test-time scaling by regularizing the easy examples.** We fine-tune a Llama-3-8B-base model on the MATH dataset with CE loss and plot the model confidence $p(y|x)$ and the factor $F(N, p(y|x))$ at 86% of the first training epoch. Both model confidence and factor are evaluated on the **unseen** data grouped by difficulty level from the MATH dataset. Model confidence decreases with increasing difficulty, whereas the regularization factor increases with problem difficulty. As a result, DCO effectively regularizes contribution from easy examples. This regularization mitigates the potential detrimental effects of overconfidence from easy examples as discussed in Appendix C.1. We use $N = 256$ corresponding to pass@256 for the plots.

**C.10    Quantifying the output diversity of models trained with DCO**

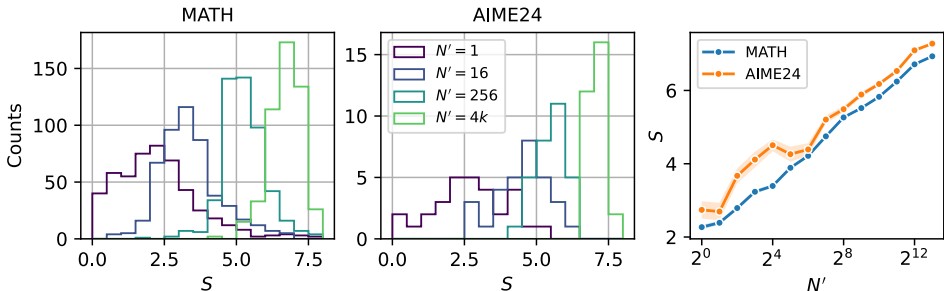

Figure 15: **DCO enhances output diversity in fine-tuned models. Increasing $N'$ leads to greater diversity in model completions.** We fine-tune a Llama-3-8B-base model on the MATH dataset to generate direct answers without explicit reasoning steps. Shannon entropy of model completions, conditioned on the input prefix, is estimated using 4096 samples per test example. **Left** and **Middle**: Histograms depicting the estimated entropy distributions for the test of MATH (left) and AIME24 (middle) respectively. Higher values of $N'$ shift the entropy distribution to the right, reflecting increased diversity of model outputs. **Right**: Mean entropy values plotted against $N'$ with standard errors of the mean. The estimated entropy increases with $N'$.

