# OpenReview forum: "Rethinking Fine-Tuning when Scaling Test-Time Compute: Limiting Confidence Improves Mathematical Reasoning"
_NeurIPS.cc/2025/Conference — NeurIPS 2025 poster_

### Official Review · Reviewer_j14W · 2025-06-09

**Clarity:** 4
**Significance:** 2
**Originality:** 3
**Rating:** 5
**Confidence:** 3

**Summary:**

This paper investigates an misalignment between the standard fine-tuning objective for large language models (cross-entropy loss) and the widely used test-time strategy of pass@N sampling. The authors demonstrate, both theoretically and empirically that training with cross-entropy loss can lead to model overconfidence. This overconfidence, while improving pass@1 accuracy somewhat paradoxically harms performance when sampling a larger number of solutions (pass@N for large N), as the model becomes less exploratory and repeatedly generates the same incorrect high-confidence answers. To address this, the paper introduces Direct Coverage Optimization, a loss function that directly optimizes the pass@N metric during training. By acting as a confidence regularizer, DCO prevents the model from becoming overconfident in its solutions. The authors show that DCO not only rescues the degrading pass@N performance but also establishes a Pareto-optimal frontier, improving results on mathematical reasoning benchmarks like MATH and theorem-proving benchmarks like MiniF2F, for both direct-answer and chain-of-thought settings.

**Questions:**

1. The results for Llama-3-70B in the appendix are encouraging as they show a similar trend to the 8B model. Could you provide more insight into the magnitude of this effect at scale? Does the performance gap between DCO and CE loss widen, shrink, or remain constant when moving from 8B to 70B? To strengthen the paper's claim of generalizability, I strongly encourage you to integrate the 70B results more centrally into the main paper, perhaps directly compare the confidence distributions and performance frontiers across scales.

2. The results show that the optimal DCO hyperparameter N' is closely tied to the test-time pass@ N. This suggests that one might need to train multiple models to be prepared for different inference-time compute budgets. What is your recommendation if the exact test-time N is not known in advance? Is there a "sweet spot" N' that provides robust performance across a wide range of test N values? Or is ensembling models trained with different N' values (as you did for theorem proving) the most practical approach?

3. While the work focuses on domains with clear verifiers, have you considered how DCO might be adapted for tasks where verification is "soft," such as using a reward model's score? How might the DCO loss function behave if the correctness signal p(y|x) is not binary (0 or 1) but a continuous score from a reward model?

**Ethical Concerns:**

["NO or VERY MINOR ethics concerns only"]

**Final Justification:**

Overall this is a good paper addressing the misalignment between cross-entropy based training and pass@N metrics during inference. The rebuttal addressed my concerns regarding the choice of N' and how the method can be used with soft verification.

**Limitations:**

yes

**Quality:**

4

**Strengths And Weaknesses:**

**Strengths**

1. The paper addresses a highly relevant and overlooked problem in LLM training for test-time scaling. The core finding that more training via standard CE loss can actively harm performance for certain inference strategies is somewhat counter-intuitive at first but makes sense on further consideration. The paper compellingly argues for a tighter co-design of training and inference protocols.

2. The paper is exceptionally well-written and clear. The authors do an excellent job of building intuition for the problem, starting with a simple two-problem example and generalizing with theoretical lemmas (4.2, 4.3). The proposed solution, DCO is elegant and directly motivated by the identified issue of overconfidence. The connection between the DCO gradient and the "overconfidence regularizer" (Figure 2b) provides a clear and satisfying explanation of why the method works.

3. The empirical validation is good. The authors test their hypothesis across multiple scenarios including direct answers, chain-of-thought reasoning on the MATH dataset, and automated theorem proving on LeanDojo. The inclusion of comparisons to both standard CE loss and a reinforcement learning baseline (GRPO) is good. The visualizations of confidence distributions (Figure 1) and performance frontiers (Fig 2) are effective and strongly support the paper's claims.

**Weaknesses**

1. The central weakness of the paper is the question of whether the core findings and the effectiveness of DCO hold at the scale of state-of-the-art models. While the authors include results for a Llama-3-70B model in the appendix (Section C.3, Figures 7 and 8) even this is not near the scale of state-of-the-art. The bulk of the results, figures, and discussion are based on the 8B model. I wonder as models become significantly larger and more capable, does this overconfidence issue persist with the same severity? It is plausible that larger models might inherently have better-calibrated uncertainty, potentially diminishing the performance gap between CE and DCO.

2. The entire premise of the pass@N strategy and the DCO objective relies on the existence of a binary verifier to determine the correctness of a generated sample. The authors acknowledge this limitation in their discussion. However, this is a major constraint that limits the applicability of the proposed method to specific domains like math, coding, and formal theorem proving. The approach is not directly usable for more open-ended tasks.

---

> ### Author Rebuttal · Authors · 2025-07-31
>
> We sincerely thank Reviewer j14W for the thorough and insightful review. We appreciate that the reviewer found the paper to be exceptionally well-written, the DCO formulation elegant, and the empirical validation strong and thorough. Below, we directly address the reviewer’s important questions and provide additional clarification and evidence to strengthen our contributions.
>
> ## Overconfidence Severity at Scale (Larger Models)
>
> We thank the reviewer for raising this important problem. Indeed, one of our motivations for investigating the Llama-3-70B model was precisely to explore whether the severity of overconfidence diminishes or intensifies as models grow larger and more capable. Our empirical findings indicate that overconfidence may become more severe at larger scales: For Llama-3-8B model, the maximum improvement of DCO frontier over CE loss happens at $N=541$ with an improvement of $26.3$%, while for Llama-3-70B model, the maximum improvement over CE loss occurs at $N=258$ with a larger improvement of $29.3$%. Interestingly, the performance gap between DCO and CE not only persists but grows slightly larger at the 70B scale. We suspect that the larger model, because it has much more parameters, may be more prone to overfitting and thus exhibit greater overconfidence under CE training. We acknowledge that this observation is based specifically on the Llama-3 series, and broader validation across more model families would further strengthen generalizability. To clarify, we are not suggesting that larger models inherently exhibit overconfidence without fine-tuning; rather, we aim to highlight a critical failure mode resulting from the misalignment between CE training and pass@N evaluation. We believe this phenomenon of training-test misalignment and the resulting overconfidence is likely a general issue.
>
> We also recognize that the pass@1 performance of the model on the benchmark is not near state-of-the-art, because we use the base model instead of an instruction-tuned model. We intentionally selected base models for two reasons: (1) to avoid potential data contamination or interference from post-training and (2) because lower pass@1 accuracy in the base models provides more meaningful room for improvement to demonstrate the effectiveness of scaling test-time compute.
>
> We will integrate these key results and comparisons in the revised paper to better address scalability concerns.
>
>
> ## Robust choices of N’ under unknown test-time compute vs. ensembling
>
> We appreciate the reviewer’s observation that the test-time $N$ might not be known at training time. To address the possibility of having “sweet-spot” choices of $N’$ for training, we evaluate the robustness of DCO to mismatches between $N′$ (the DCO objective parameter) and $N$ (the $N$ used to evaluate pass@N at test time). We define a choice of $N’$ to be optimal for a given test-time $N$, if its pass@N performance is within 1% of the DCO frontier at that $N$ (see Figure 2a for the frontier and individual DCO curves). Based on experiments fine-tuning Llama-3-8B on direct-answer generation for the MATH dataset, we observe:
>
> - $N’=1$ is optimal for $1\leq N\leq 3$.
> - $N’=16$ is optimal between $2\leq N\leq 20$.
> - $N’=256$ is optimal between $180\leq N\leq 385$.
> - $N’=4096$ is optimal for $N\geq 1151$.
>
> These results indicate that while matching $N’=N$ is a standard and intuitive approach, a fixed choice of $N’$ can result in near-optimal performance for a range of test-time $N$. Selecting a few strategic $N’$ values can thus yield near-optimal performance across broad ranges.
>
> We also believe that ensemble may be beneficial for certain use cases such as theorem proving. As highlighted by the reviewer, an ensemble of models trained with different $N_{eff}$ values can actually surpass the performance of a single model when using larger $N$ in theorem proving. Thus, training multiple specialized models can be an advantageous strategy rather than purely a limitation. Of course, both the robustness of DCO to the test-time $N$ and the benefits due to ensembling models at various choices of $N’$ are likely task-dependent.
>
> Regarding the question of whether a single model could effectively handle multiple inference budgets at test time, we would like to suggest a potential strategy: explicitly conditioning the model on the target sampling budget $N$ during training and testing (i. e. learning the distribution of $p(y|x,N)$), thus enabling a single model to flexibly adapt to various inference budgets. While exploring this direction in depth remains beyond the scope of our current work, we acknowledge it as a valuable direction for future research.
>
>
> ## Settings with “soft” verification
>
> We agree with the reviewer regarding the importance of exploring extensions of DCO to domains where verification is not strictly binary but rather “soft”. We acknowledge that restricting the scope to verifiable problems is one of the limitations as we stated at the end of our paper. Extending DCO to handle continuous verification signals represents a promising direction for future research. Given the substantial density and content of our current manuscript, we leave such extensions to future work and hope the reviewer could understand.
>
> We want to thank the reviewer again for their thoughtful comments and constructive suggestions. We remain eager to address any additional questions or concerns the reviewer might have.

---

> > ### Author Response · Authors · 2025-08-06
> >
> > Dear Reviewer j14W,
> >
> > We hope this message finds you well! We sincerely thank you for your thorough and positive evaluation of our submission. We are especially grateful for your recognition of the clarity of our presentation, the elegance of the proposed DCO method, and the strength of our empirical validation.
> >
> > We wanted to kindly follow up regarding our responses to your important questions about the scalability of the identified overconfidence issue and the robustness of our approach across different inference-time budgets. In response to your insightful feedback, we have further highlighted results for the Llama-3-70B model, demonstrating that the issue of overconfidence persists or even becomes more pronounced at larger scales, reinforcing our conclusions. Additionally, we have discussed practical considerations for selecting optimal $N'$ values and potential strategies, such as model ensembling or conditioning on inference budgets, to manage uncertainty about the exact inference-time compute available.
> >
> > If there remain any unresolved points or if further clarifications are needed, please let us know! We would be delighted to provide additional details.
> >
> > Thank you again for your valuable insights, thoughtful questions, and constructive suggestions, which have greatly contributed to improving our paper.

---

> > > ### Comment · Reviewer_j14W · 2025-08-07
> > >
> > > Dear authors,
> > >
> > > Thank you for the detailed response - you have clarified the questions I had. The results regarding N' are interesting and it would be useful to include in the paper.
> > >
> > > Regards,
> > > j14W

---

> > > > ### Author Response · Authors · 2025-08-07
> > > >
> > > > Dear Reviewer j14W,
> > > >
> > > > Thank you very much for your kind response and helpful feedback. We are delighted to hear that our clarifications addressed your questions effectively. We appreciate your suggestion regarding the results on $N'$, and we will certainly incorporate these findings into the revised manuscript to further strengthen our work.
> > > >
> > > > Once again, thank you for your thoughtful insights and valuable contributions to improving our paper.

---

### Official Review · Reviewer_52g7 · 2025-06-30

**Clarity:** 3
**Significance:** 3
**Originality:** 3
**Rating:** 4
**Confidence:** 3

**Summary:**

The paper investigates the misalignment between cross-entropy (CE) fine-tuning and test-time compute scaling (e.g., pass@N). It shows that prolonged CE training induces overconfidence, degrading pass@N performance despite improving pass@1. The authors attribute this to a tradeoff between exploitation (high confidence) and exploration (diverse sampling). To realign training with pass@N, they propose Direct Coverage Optimization (DCO), a loss that limits model confidence via a regularization term attenuating gradients for high-confidence predictions. Experiments on MATH (direct answers/CoT) and LeanDojo (theorem proving) demonstrate that DCO rescues pass@N scaling and achieves Pareto-optimal frontiers. Extensions to step-wise DCO for proof trees and approximate DCO for CoT further validate the framework.

**Questions:**

See the Weakness.

**Ethical Concerns:**

["NO or VERY MINOR ethics concerns only"]

**Final Justification:**

The detailed response solves my concern.

**Limitations:**

yes

**Paper Formatting Concerns:**

N/A.

**Quality:**

3

**Strengths And Weaknesses:**

Strength

- Novel Insight: Identifies a counterintuitive phenomenon—longer CE training harms pass@N performance—and traces it to overconfidence, supported by theory (Lemmas 4.2–4.3) and empirical validation (Table 1, Figure 1).

- Practical Impact: The ensemble of step-wise DCO models (Table 2) boosts theorem-proving success by +5.2% (MathLib) and +4.1% (MiniF2F) over CE with matched compute, highlighting real-world utility.

- Theoretical Depth: Lemmas 4.2–4.3 formally characterize the confidence-exploration tradeoff, and Appendix A provides complete proofs.

Weakness

- Approximate DCO for CoT requires online Monte Carlo estimation, increasing compute by 10× vs. standard fine-tuning.

- The paper links overconfidence to reduced pass@N performance but does not quantify how DCO affects output diversity.

---

> ### Author Rebuttal · Authors · 2025-07-31
>
> We thank the reviewer for their thorough review and insightful feedback and suggestions. We are pleased that the reviewer recognizes the novelty and practical relevance of our findings and appreciates the depth of our theoretical analysis. Below, we directly address the reviewer's concerns and provide additional evidence and clarifications.
>
> ## The effect of DCO on output diversity
> We thank the reviewer for suggesting this point. We agree with the reviewer that the effect of the DCO objective on output diversity is important to understand. To quantify this, we estimated the mean Shannon entropy of the model outputs using Monte Carlo sampling with 4096 samples per problem. Specifically, we fine-tuned a Llama-3-8B-base model on the MATH dataset to generate direct answers using DCO losses with different values of N’ and compute the mean Shannon entropy of the model distribution on the test set of MATH and AIME24. We will explicitly incorporate these diversity results into our revised paper. We summarize the results in the table below:
>
> **Table 1**.  Mean and standard deviation of Shannon entropy for models trained with DCO of varying N’ values.
> | Dataset | $N’=1$ (CE loss)  | $N’=8$  | $N’=64$  | $N’=512$  | $N’=4096$ |
> |----------|----------|----------|----------|----------|----------|
> | MATH   | $2.27\pm1.43$    | $3.23\pm1.16$    | $4.22\pm0.91$     | $5.52\pm0.71$     |  $6.71\pm0.58$  |
> | AIME24    | $2.74\pm1.27$    | $4.11\pm1.17$    | $4.39\pm0.91$     | $5.89\pm0.41$     |  $7.10\pm0.22$  |
>
> The table clearly demonstrates that the entropy consistently increases with larger $N’$. This indicates that models trained with higher $N’$ indeed produce more diverse outputs, thus directly confirming that DCO naturally encourages diversity. Intuitively, superior pass@N performance at larger $N$ requires distributing probability mass across multiple candidate solutions, reducing overconfidence and enhancing coverage.
>
> ## Computational overhead in the CoT setting
>
> We agree with the reviewer regarding the significant computational overhead involved in Monte Carlo estimation in CoT settings. While acknowledging this limitation, we briefly note here that, although this overhead is significant relative to standard SFT, costly online inference is also required in RL frameworks for CoT training, as discussed in Appendix B. Importantly, practitioners can significantly reduce training costs by using fewer Monte Carlo samples, trading precision in estimating success probability for increased computational efficiency.
>
>
> We thank the review again for their thoughtful questions and constructive feedback. We hope our detailed responses address all concerns, and we remain available to provide further clarifications if needed.

---

### Official Review · Reviewer_asva · 2025-07-01

**Clarity:** 3
**Significance:** 4
**Originality:** 3
**Rating:** 4
**Confidence:** 3

**Summary:**

This paper focus on the problem that after few epochs of training, while the pass@1 result of model increases, pass@N decreases. It claims a mismatch between how we train LLMs (with cross-entropy loss that optimizes single best output) and how we test them (by pass@N), and the standard fine-tuning makes the model too confident at one guess and hurts performance when we sample many times. They propose a new loss objective, Direct Coverage Optimization (DCO), which directly maximizes the chance of getting at least one correct answer in N samples, showing gains in  pass@N on math problems and theorem proving benchmarks.

**Questions:**

In the paper, the authors train with N’ = N to match the test-time sampling budget. But retraining a separate model for every target N can be very costly. How robust is a model trained for one N’ when tested at different budgets (e.g., training at N’=16 but testing at N=64 or N=4)? Have you explored stragety to support multiple budgets without full retraining for each?

**Ethical Concerns:**

["NO or VERY MINOR ethics concerns only"]

**Limitations:**

yes

**Quality:**

4

**Strengths And Weaknesses:**

Strengths:
 - The paper provides a insightful observation and points out a real gap between training and inference and explains why it matters, with clear proofs show why cross-entropy leads to overconfidence and how DCO fixes it.
 - The performance of DCO and its variants are good on MATH, AIME, MiniF2F, and Mathlib, cross multiple domains.
 - The method applies to both step-wise proof search and chain-of-thought, showing robustness.

Weaknesses:
 - Training with DCO may need more sampling and GPU time, which is acceptable.

---

> ### Author Rebuttal · Authors · 2025-07-31
>
> We sincerely thank Reviewer asva for their thorough and insightful feedback. We are pleased that the reviewer found our identification of the gap between training and inference insightful, and recognized the robustness and strong empirical performance of our proposed DCO. Below, we address the reviewer's main concerns and questions directly, providing additional clarifications and evidence to strengthen our contribution.
>
> ## Robustness of DCO
>
> We appreciate the reviewer’s observation that training a separate model for each choice of $N'$ is computationally expensive. Indeed, the central theme for this paper is that we cannot simply train a single model to be optimal for every $N$ without conditioning on $N$ at test time.
>
> The reviewer raises a question about the robustness of DCO to mismatches between $N'$ (the DCO objective parameter) and $N$ (the $N$ used to evaluate pass@N at test time). We would like to remind the reviewer that there is an example plot in Figure 6, where we highlight and show how robust a model trained with a specific $N=256$ is with respect to the Pareto frontier. To further investigate the robustness, we define an $N'$ as optimal for a given $N$ if its pass@N performance is within 1% of the DCO frontier at that $N$ (see Figure 2a). Based on experiments fine-tuning Llama-3-8B on direct-answer generation for the MATH dataset, we observe:
>
> - $N’=1$ is optimal for $1\leq N\leq 3$.
> - $N’=16$ is optimal between $2\leq N\leq 20$.
> - $N’=256$ is optimal between $180\leq N\leq 385$.
> - $N’=4096$ is optimal for $N\geq 1151$.
>
> These results strongly indicate that while matching $N'=N$ is a standard and intuitive approach, a fixed choice of $N'$ can result in near-frontier performance for a range of test-time $N$. Selecting a few strategic $N'$ values can yield near-optimal performance across broad ranges, reducing computational overhead.
>
>
> Moreover, as demonstrated in our theorem proving results, an ensemble of models trained with different $N'$ values can actually surpass the performance of a single model using larger $N$. Thus, training multiple models and forming an ensemble can be an advantageous strategy rather than purely a limitation.
>
>
> ## Strategies for Supporting Multiple Test-time Budgets
>
> The reviewer raises an important question regarding potential strategies to support multiple test-time budgets without complete retraining. Indeed, a promising approach is to explicitly condition the model on $N$ during training and testing (i. e.  $p(y|x,N)$), enabling a single model to flexibly adapt to various inference budgets. While exploring this direction in depth remains beyond the scope of our current work, we acknowledge it as a valuable direction for future research.
>
>
>
> ## Computational overhead in the CoT setting
> We agree with the reviewer regarding the significant computational overhead involved in Monte Carlo estimation in CoT settings. While acknowledging this limitation, we briefly note here that, although this overhead is significant relative to standard SFT, costly online inference is also required in RL frameworks for CoT training, as discussed in Appendix B. Importantly, practitioners can significantly reduce training costs by using fewer Monte Carlo samples ($N_{MC}$), trading precision in estimating success probability for increased computational efficiency.
>
>
> We thank the reviewer again for their valuable suggestions and insightful questions, which help improve our paper substantially. We hope our responses clearly address their concerns and further strengthen their confidence in our findings. Please don’t hesitate to reach out if anything remains unclear or if additional questions arise.

---

> > ### Author Response · Authors · 2025-08-06
> >
> > Dear Reviewer asva,
> >
> > We hope you are doing well! We greatly appreciate your insightful feedback and positive evaluation of our submission, especially your recognition of the identified gap between training and inference and the robustness of our proposed DCO method.
> >
> > We wanted to kindly follow up to ensure that our rebuttal has fully addressed your questions, particularly regarding the robustness of DCO to mismatched training and testing sampling budgets and strategies to support multiple test-time budgets without extensive retraining. As outlined in our response, we've demonstrated that a carefully selected fixed $N'$ can achieve near-frontier performance across a range of test-time $N$. We also acknowledge and propose explicitly conditioning the model on $N$ as an exciting avenue for future work.
> >
> > If there remain any unresolved points or further clarifications you might require, please let us know. We would be very happy to provide additional details.
> >
> > We would greatly appreciate it if you could revisit your evaluation in light of our responses and potentially update your score to reflect the improvements we've incorporated.
> >
> > Thank you once again for your valuable insights and consideration, which substantially contribute to the quality and clarity of our paper.

---

> > ### Comment · Reviewer_asva · 2025-08-08
> >
> > Thanks for your clarification and it makes sense. I will keep my rating.

---

### Official Review · Reviewer_4Nd6 · 2025-07-04

**Clarity:** 3
**Significance:** 3
**Originality:** 3
**Rating:** 4
**Confidence:** 4

**Summary:**

The paper investigates a subtle but important mismatch between how LLMs are fine-tuned and how they are later used with heavy test-time sampling (e.g. pass@N). The authors first show that training with standard cross-entropy (CE) pushes the model toward over-confident, low-diversity distributions that hurt coverage when many samples are drawn. On MATH, for instance, pass@256 accuracy falls from 65% after one epoch to 45% after four epochs despite pass@1 steadily improving. They trace the effect to a theory-backed exploit/explore trade-off and introduce Direct Coverage Optimization (DCO), a simple loss that directly maximizes the probability a correct answer appears in N samples while automatically regularizing over-confidence. Results on math are quite promising.

**Questions:**

- Regarding N: Could you provide a rule-of-thumb (perhaps tied to available inference budget) to pick these hyper-parameters without sweeping? A calibration plot would make adoption easier.
 - The epsilon part is a bit sketchy. Is it really necessary? What's the rule of thumb for that?
 - As sequence length gets longer, the prob of the sequence naturally gets lower. How does that affect things?

**Ethical Concerns:**

["NO or VERY MINOR ethics concerns only"]

**Limitations:**

yes

**Quality:**

3

**Strengths And Weaknesses:**

Strengths:
 - Solid theoretical analysis explaining why CE induces over-confidence;
 - Writing is clear; figures/tables make trade-offs visually clear
 - Method is simple and easy to implement
 - Topic very relevant today

Weakness:
 - Related work on inference-aware fine-tuning (e.g. BonBon, BOND) is acknowledged but could be contrasted more deeply, especially on diversity vs. confidence control
 - There are many previous loss functions that try to regularize cross entropy to avoid over confidence. Focal loss, which is commonly used in computer vision, is one such example. Could you discuss (or maybe even compare to) those losses?

---

> ### Author Rebuttal · Authors · 2025-07-31
>
> We sincerely thank the reviewer for their thorough and insightful comments. We are encouraged that the reviewer found the theoretical analysis solid, our writing clear, and our topic timely and relevant. Below, we directly address the reviewer's concerns and questions, with additional empirical results and clarifications to strengthen our contributions.
> ## Comparison to Focal Loss
> The reviewer correctly notes that various works have also proposed loss functions aiming to reduce overconfidence in the cross entropy (CE) loss. However, we would like to emphasize that the overconfidence we identified arises specifically from a train-test mismatch in sampling strategies. While there are other methods/loss functions to alleviate overconfidence, our DCO loss is directly designed to address this misalignment.
> To concretely demonstrate this difference, we followed the reviewer’s suggestion and conducted a thorough comparison with Focal Loss (FL) [Lin et al., 2017], $\ell^\gamma_\text{FL}(x, y) = -(1-\hat{p}(y \mid x))^\gamma \log \hat{p}(y \mid x)$, optimizing over its hyperparameter $\gamma$ for each value of $N$. Our results are as follows:
>
>
> **Table 1**. We fine-tune a Llama-3-8B-base model on the MATH dataset to generate direct answers with CE loss, FL with different $\gamma$ and the DCO losses with different $N$, and test the model on the test set of MATH. We have reported pass@N performance for the CE baseline and DCO frontier (as plotted in Figure 2a). Here, we include the FL frontier, which selects the best performing $\gamma$ for each $N$. The FL frontier consistently outperforms the CE baseline but underperforms the DCO frontier across all values of $N$. This further highlights the advantage of DCO in improving coverage under heavy sampling.
> | | N=1 | N=8 | N=64 | N=512 | N=1024 | N=2048 | N=4096 |
> |----------|----------|----------|----------|----------|----------|----------|----------|
> | CE   | **0.07**     | 0.18     | 0.33     | 0.49     |  0.54   |  0.59     |   0.63    |
> | Focal Loss frontier    | **0.07** | 0.22  | 0.44 | 0.63  | 0.68  | 0.72  | 0.75  |
> | DCO frontier    | **0.07**     | **0.24**     | **0.54**     | **0.75**     |**0.80** | **0.84**    |  **0.87**     |
>
> **Table 2**. Same as Table 1, but test on the out-of-distribution AIME24.
>
> |                      | N=1  | N=8  | N=64 | N=512 | N=1024 | N=2048 | N=4096 |
> |----------------------|------|------|------|-------|--------|--------|--------|
> | CE                   | **0.02** | 0.04 | 0.09 | 0.19  | 0.23   | 0.29   | 0.36   |
> | Focal Loss frontier  | **0.02** | 0.04 | **0.12** | 0.24  | 0.33   | 0.45   | 0.55   |
> | DCO frontier         | **0.02** | **0.04** | 0.11 | **0.48** | **0.64** | **0.79** | **0.89** |
>
> These results clearly show that while heuristically designed objectives like FL can indeed outperform vanilla CE loss, DCO loss consistently achieves significantly superior performance, particularly at larger $N$ values critical for practical applications. We will include these comparisons explicitly in the revised paper. Specifically, we will add a corresponding plot illustrating that the FL frontier consistently lies above the CE baseline but below the DCO frontier across nearly all values of $N$ on both in-distribution MATH test set and out-of-distribution AIME24.
>
> We fully agree that providing clear guidelines for choosing $N’$ greatly enhances the practicality of our approach, given that training a separate model for each choice of $N'$ is computationally expensive. The main argument of this paper is that one should align the training objective with the test-time strategy. That said, we study the calibration of the DCO objective and find that DCO can be robust to some extent to mismatches between $N'$ (the DCO objective parameter) and $N$ (the $N$ used to evaluate pass@N at test time). We define an $N'$ as optimal for a given $N$ if its pass@N performance is within 1% of the DCO frontier at that $N$ (see Figure 2a). Based on experiments fine-tuning Llama-3-8B on direct-answer generation for the MATH dataset, we observe:
> - $N’=1$ is optimal for $1\leq N\leq 3$.
> - $N’=16$ is optimal between $2\leq N\leq 20$.
> - $N’=256$ is optimal between $180\leq N\leq 385$.
> - $N’=4096$ is optimal for $N\geq 1151$.
>
> We also remind the reviewer that we proposed a rule of thumb for selecting $N_{\text{eff}}$ in the case of theorem proving (see the paragraph starting at line 269). If one has an estimate of task difficulty (e.g., proof length $k$), we suggest using $N_{\text{eff}} = N^{-1/k}$, where $N$ is the inference-time sampling budget.
> Additionally, if an inference-time budget (in FLOPs or wall-clock time) is available, a practical strategy is to estimate the distribution of cost per sample and then choose the largest $N'$ such that the expected total cost remains within budget.
>
>
> ## Clarification of the role of $\epsilon$ in the lemmas
> We thank the reviewer for raising this important point. We clarify that $\epsilon$ captures uncertainty regarding exact probability values for lower-ranked terms. The aim of Lemma 4.2 is to establish an upper bound for the model confidence $\hat{p_1}$* associated with the highest-ranked answer under the optimal policy that maximizes coverage. The precise value of $\hat{p_1}$* inherently depends on all probabilities $p_{i}$. The introduction of the $\epsilon$ term explicitly acknowledges uncertainty regarding the exact values of probabilities $p_{i}$ for indices $i>k$, capturing our incomplete knowledge of these lower-ranked terms. Thus, $\epsilon$ is a theoretically motivated component ensuring robustness rather than a practically burdensome hyperparameter.
>
> ## Comparison to other inference-aware tuning methods
> We appreciate the suggestion to expand our discussion regarding BonBon and BOND. Both methods elegantly address the problem of distilling the Best-of-$N$ (BoN) policy, thereby avoiding the cost of $N$ inference passes at test-time, closely related to RLHF objectives. However, their work focuses on improving the pass@1 performance from the BoN policy, while we focus on directly improving the pass@N performance. We will incorporate these distinctions clearly in our revised related-work section, highlighting the complementary contexts in which each method applies.
> ## Effect of sequence length on the probability
> We recognize the reviewer’s concern regarding longer sequences naturally receiving lower probabilities. Importantly, our method is designed such that sequence length does not inherently penalize coverage:
> In direct-answer tasks, DCO optimizes directly for the correct final answer, independent of sequence length.
> For formal math, we apply DCO at the step-level granularity, avoiding any length-related bias.
> In Chain-of-Thought scenarios, our method marginalizes over reasoning traces, ensuring no direct penalization of longer sequences.
> Thus, sequence length inherently does not disadvantage our proposed approach.
>
>
> We greatly appreciate Reviewer 4Nd6's valuable feedback, which significantly strengthens our paper. We hope our detailed clarifications and additional experiments address all concerns and increase the reviewer’s confidence in our work. We would be happy to clarify any points that are still unclear or any additional questions.
>
> Reference:
>
> [Lin et al., 2017] Lin, Tsung-Yi, et al. "Focal loss for dense object detection." Proceedings of the IEEE international conference on computer vision. 2017.

---

### Decision · Program_Chairs · 2025-09-17

**Decision:**

Accept (poster)

**Comment:**

In this paper, the authors note that large language models are often evaluated on pass@k evaluations (i.e. whether one sample is correct from k samples conditioned on a prompt) but they are trained on pass@1.  They observe that models trained on standard objectives actually degrade in performance on pass@k.  As such, they propose a new training loss they call Direct Coverage Optimization (DCO).  They show empirically that DCO improves on pass@k evaluations compared to cross-entropy and the recent GRPO method.  They also show theoretically why pass@k performance degrades with existing strategies.

Overall the reviews are borderline leaning towards accept (4, 4, 4, 5).  The reviewers seemed to find the paper "exceptionally" well written and clear.  They found the work well-motivated, topically "very relevant" and theoretically "solid".  They noted that the method presented was simple and seemed empirically convincing.  The reviewers who gave scores of 4, noted very little in terms of weaknesses and overal seemed positive.  Unfortunately, that didn't give the authors a significant amount to address and, in-addition, those reviewers didn't seem to engage meaningfully in discussion despite what seems like an in-depth response.  Two of these reviewers noted that the proposed algorithm required additional sampling and thus was computationally less efficient (however, GRPO does as well).  The reviewer that recommended accept gave a thorough review and engaged with the authors.  Given the lack of any significant concerns raised by some of the reviewers, I would deduce that their scores are perhaps lower than they should be.  Therefore, the recommendation is to accept (i.e. up-weighing the 5 score).

The Accept review noted that there remains a question whether the results generalize to the current largest SOTA models.  Naturally, that is difficult to empirically verify, but seems like an important potential limitation.